# TRIM: Hybrid Inference via Targeted Stepwise Routing in Multi-Step Reasoning Tasks

**Vansh Kapoor**[1†]**, Aman Gupta**[2]**, Hao Chen**[2]**, Anurag Beniwal**[2]**, Jing Huang**[2]**, Aviral Kumar**[1]
[1]Carnegie Mellon University        [2]Amazon

## Abstract

Multi-step reasoning tasks like mathematical problem solving are vulnerable to cascading failures where a single incorrect step leads to complete solution breakdown. Current LLM routing methods assign entire queries to one model, treating all reasoning steps as equal. We propose TRIM (Targeted routing in multi-step reasoning tasks), which routes only critical steps–those likely to derail the solution–to larger models while letting smaller models handle routine continuations. Our key insight is that targeted step-level interventions can fundamentally transform inference efficiency by confining expensive calls to precisely those steps where stronger models prevent cascading errors. TRIM operates at the step-level: it uses process reward models to identify erroneous steps and makes routing decisions based on step-level uncertainty and budget constraints. We develop several routing strategies within TRIM, ranging from a simple threshold-based policy to more expressive policies that reason about long-horizon accuracy-cost trade-offs and uncertainty in step-level correctness estimates. On MATH-500, even the simplest thresholding strategy surpasses prior routing methods with $5\times$ higher cost efficiency, while more advanced policies match the strong, expensive model's performance using $80\%$ fewer expensive model tokens. On harder benchmarks such as AIME, TRIM achieves up to $6\times$ higher cost efficiency. All methods generalize effectively across math reasoning tasks, demonstrating that step-level difficulty represents fundamental characteristics of reasoning.

## 1 Introduction

The rapid progress in large language models (LLMs) has led to a diverse ecosystem of models, spanning a wide spectrum of sizes, capabilities, and computational demands. Larger models typically achieve stronger performance but incur substantial serving costs, rendering them impractical for many routine applications. In contrast, smaller models are more affordable to deploy but often produce lower-quality responses. This trade-off poses a fundamental dilemma for the practical deployment of LLMs: routing all queries to the largest available model ensures high-quality outputs but is prohibitively expensive, whereas relying solely on smaller models reduces serving costs at the expense of degraded response quality, especially on challenging queries.

Contemporary routing strategies attempt to mitigate this dilemma by assigning each query to a single model, which is then responsible for the *entire generation*. However, not all tokens in an LLM's response are equally challenging to generate: some represent critical decision points that can dramatically alter the solution path (Setlur et al., 2024a; Wang et al., 2025; Qu et al., 2025), while others are routine continuations that are easier to generate. By committing to a larger LLM over a smaller one for a given query, existing routing methods implicitly assume that the stronger model's intervention is equally necessary at every token to produce a high-quality response. This inefficiency is particularly pronounced in multi-step reasoning tasks such as step-by-step reasoning or code generation, where mistakes early on can snowball into a complete failure (Zhang et al., 2024). It is precisely at these erroneous steps that the intervention of a stronger model is most valuable. Yet, contemporary routing methods often incur substantial inefficiency by defaulting to full generations from the larger model, even when targeted interventions at these erroneous steps would suffice. To address this inefficiency, we introduce TRIM-**T**argeted Stepwise **R**outing for **I**nference in **M**ulti-step

---

[†]Work primarily conducted during an internship at Amazon. Corresponding author: `vanshk@cs.cmu.edu`

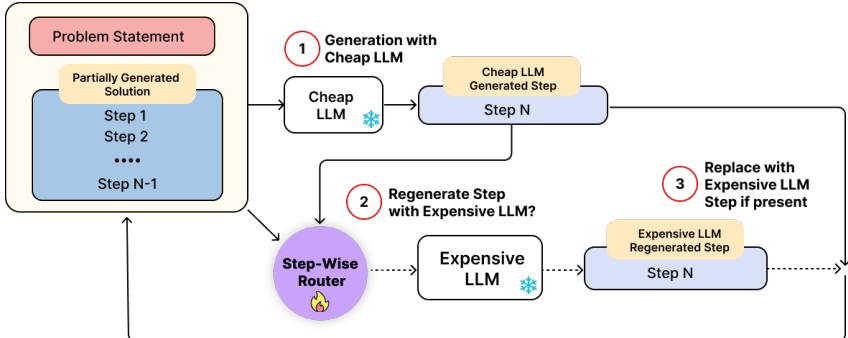

Figure 1: Schematic Overview of a Two-Model Setup for TRIM

Reasoning Tasks: an approach that selectively routes only the most critical steps to larger LLMs. Unlike prior prompt-level routers, TRIM operates at the granularity of individual reasoning steps, generating solutions one step at a time. As illustrated in Figure 1, a stepwise router evaluates each intermediate step as it is generated and decides whether to accept the small model's output or to regenerate that specific step with a larger model. TRIM ensures that interventions occur only when necessary at individual steps, rather than handing over the entire remaining solution to the larger model. This step-by-step generation process with targeted intervention enables TRIM to achieve efficiency by confining expensive calls to precisely those steps where intervention prevents cascading errors while allowing the smaller model to handle routine continuations. This approach reduces the primary cost bottleneck in inference: the number of tokens generated by the expensive, larger LLM.

Building on this framework, we design multiple strategies for stepwise routing, each tailored to different computational and informational constraints: a simple thresholding policy that uses step-level scores to identify erroneous steps, two RL-trained policies that reason about long-horizon trade-offs between accuracy and cost (one using full sequential features, another using aggregated statistics), and a Partially Observable Markov Decision Process (POMDP) based approach that accounts for the inherent uncertainty in step-level correctness estimates while enabling efficient policy recomputation across different cost budgets. Our cost metric focuses on the number of tokens generated by the expensive model, as prefill costs can be amortized through parallel decoding strategies (Leviathan et al., 2023; Cai et al., 2024), whereas generation tokens impose unavoidable sequential costs. We evaluate our approach on diverse mathematical reasoning benchmarks, including MATH (Hendrycks et al., 2021), Olympiad-Bench (He et al., 2024), and AIME (Di Zhang, 2025). On MATH-500, our basic thresholding policy is 5× more cost-effective than existing methods, while our trained RL and POMDP policies achieve the performance of the expensive LLM using only $20\%$ of the expensive tokens. On the more challenging AIME benchmark, these policies achieve $3.17\times$ and $6.33\times$ higher cost efficiency, respectively. Moreover, routing policies trained on one dataset generalize strongly to others: the RL-trained policy attains up to $11.68\times$ higher cost efficiency when evaluated on OlympiadBench, despite being trained only on AIME. Our main contributions are: **(1)** We provide systematic evidence showing that a small number of targeted step-level interventions can substantially enhance the efficiency of multi-step reasoning. **(2)** We demonstrate that this insight can be used to instantiate practical frameworks that are amenable to several routing policies, from simple thresholding policies that surpass existing query-level routing methods to advanced RL-trained and POMDP-based approaches that achieve competitive performance with oracle routers having perfect task knowledge, all while using significantly fewer expensive model tokens. Our routing policies can be trained effectively with limited supervision data while achieving robust performance across diverse cost budgets, making the approach practical for real-world deployment scenarios. **(3)** Finally, we demonstrate robust generalization of the routing strategy across datasets, with methods trained on AIME delivering strong efficiency gains on OlympiadBench and Minerva Math, suggesting that step-level difficulty patterns capture transferable structure across related reasoning benchmarks for a given base model, rather than being tightly coupled to a specific dataset.

## 2 RELATED WORK

The trade-off between model performance and computational cost has become increasingly salient as large language models grow in size and capability. Our work on targeted stepwise routing builds upon

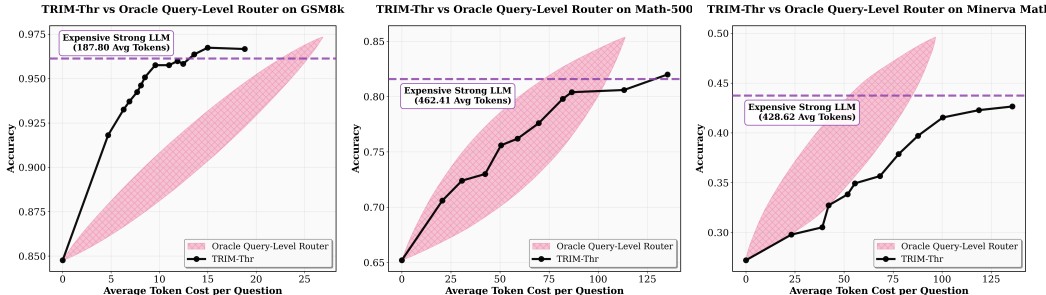

Figure 2: **Comparison of task performance–cost trade-offs** for Qwen2.5-3B-Instruct ($M_w$) and Claude 3.7 Sonnet ($M_s$), under the myopic thresholding policy (with Qwen2.5-Math-PRM-7B) versus the Idealized Oracle Query-level Router, across multiple math benchmarks. The Oracle Router is evaluated by incrementally varying the number of queries routed to $M_s$, selecting those solvable by $M_s$ but not by $M_w$. Since multiple query subsets can achieve the same accuracy with different token costs, the oracle's performance–cost curve forms a shaded region rather than a single line, reflecting the full trade-off frontier.

prior work from several areas: LLM routing strategies, process supervision and step-level verification, and multi-step reasoning optimizations.

**LLM routing and model selection.** Traditional routing approaches operate at the query level, assigning entire queries to a single model based on estimated difficulty or uncertainty. Under this query-level paradigm, Hybrid-LLM (Ding et al., 2024) frames routing as a classification task using BERT-style encoders, while Zooter (Lu et al., 2023) employs reward-guided training for normalized reward prediction. RouteLLM (Ong et al., 2024) learns routing directly from preference data, and AutoMix (Aggarwal et al., 2023) formulates query-level routing as a POMDP with self-verification for difficulty estimation. Recent advances extend these approaches in complementary directions. BEST-Route (Ding et al., 2025) augments routing with adaptive test-time compute, jointly selecting a model and number of samples to generate per query (best-of-$n$) based on query difficulty and quality thresholds However, all these methods assume uniform difficulty across generation steps, leading to inefficient resource allocation in multi-step reasoning tasks where difficulty varies significantly across steps. In such settings, errors are often localized, and correcting a few critical steps is often sufficient to steer the solution back onto a successful path. In contrast, TRIM departs from query-level routing by operating at the granularity of individual reasoning steps, enabling targeted interventions precisely where stronger models are most valuable while avoiding unnecessary escalation elsewhere.

**Process supervision and step-level verification.** Process reward models (PRMs) have emerged as a powerful technique for evaluating intermediate reasoning steps. Human-in-the-loop step verification method (Lightman et al., 2023) showed that process supervision significantly outperforms outcome supervision on math reasoning tasks, motivating scalable automated alternatives. Subsequent work developed automated process supervision (Luo et al., 2024) and stepwise verification without human annotations (Wang et al., 2023). Recent studies further demonstrate that strategic computation allocation (Hwang et al., 2024; Snell et al., 2024; Setlur et al., 2024b) can significantly improve mathematical reasoning efficiency. While these works primarily use PRMs for candidate selection or exploration shaping, our work instead uses PRM scores to inform routing decisions during generation.

**Multi-step reasoning and test-time compute.** Recent work has highlighted the importance of strategic computation allocation in multi-step reasoning. Research on "forking paths" (Bigelow et al., 2025) and (Wang et al., 2025) demonstrates that certain tokens represent critical decision points that dramatically alter solution trajectories. Similarly, work on reinforcement learning for mathematical reasoning (Setlur et al., 2024a; DeepSeek-AI et al., 2025; Yang et al., 2025; Team et al., 2025; Yu et al., 2025; Lin et al., 2024) and optimizing test-time compute (Snell et al., 2024; Qu et al., 2025) shows that targeted interventions at crucial steps can be more effective than uniform computation increases. The observation that high-entropy minority tokens drive effective learning (Wang et al., 2025) further supports the notion that not all generation steps are equally important.

**Speculative and parallel decoding.** Our approach shares some similarities with speculative decoding (Leviathan et al., 2023) and parallel inference strategies (Cai et al., 2024), which also involve multiple models collaborating during generation. However, these methods focus on acceleration

through draft-and-verify paradigms, while our work targets the quality-cost trade-off in multi-step reasoning through selective model escalation. Reward-Guided SD (Liao et al., 2025) and SpecReason (Pan et al., 2025) are closer in spirit in that they use step-level signals to accept or regenerate intermediate generations (RSD via PRMs, SpecReason via expensive model-generated scores), but they differ fundamentally in objective. Both operate in a fixed, high-budget regime and are optimized primarily for latency reduction with marginal accuracy gains, whereas TRIM explicitly optimizes accuracy under a constrained cost budget by limiting strong-model token usage. Although RSD and SpecReason can be adapted to routing by varying acceptance thresholds, such adaptations remain inherently myopic: decisions depend only on the current step's score and do not account for noise in correctness estimates or long-horizon budget allocation. By contrast, TRIM is more general and complementary: it subsumes threshold-based rules while enabling more expressive policies that reason about trajectory-level progress, uncertainty in step-level correctness, and expected future benefit of intervention, leading to more robust and efficient compute allocation in multi-step reasoning.

## 3   PRELIMINARIES AND PROBLEM STATEMENT

We define **stepwise routing** within a multi-step reasoning task as a sequential decision process. The objective is to derive a routing policy that, at each step of generation, decides whether to (i) accept the output of a cheap language model, or (ii) re-generate the step using a more capable but expensive LLM, thereby incurring an additional per-token cost. This policy must balance the trade-off between maximizing the task reward of the final solution and minimizing the serving cost, defined as the number of tokens generated from the expensive, stronger LLM. We formalize this problem below.

**Problem Formulation.** A multi-step reasoning task is specified by a query $q \in \mathcal{Q}$ and a sequence of reasoning steps

$$\mathbf{y}_{1:N} = (q, y_1, y_2, \ldots, y_N) \in \mathcal{Y}_N,$$

where $y_i$ denotes the $i$-th reasoning step. For our purposes, we assume these steps are delimited by double newlines in the generated text. At time-step $t$, the current prefix is denoted $\mathbf{y}_{1:t} = (q, y_1, \ldots, y_t) \in \mathcal{Y}_t$, where $\mathcal{Y}_t$ denotes the set of all prefixes of length $t$. Within $\mathcal{Y}_t$, we distinguish two disjoint subsets: (1) $\mathcal{P}_t \subseteq \mathcal{Y}_t$, the set of *incomplete prefixes* (partial answers) that can be extended further; (2) $\mathcal{C}_t \subseteq \mathcal{Y}_t$, the set of *completed (terminated) answers* at step $t$.

Let $\mathcal{M}$ denote a set of language models, where each model $M \in \mathcal{M}$ maps a partial reasoning trace $\mathbf{y}_{1:t} \in \mathcal{P}_t$ to the next step $\mathbf{y}_{1:t+1} \in \mathcal{Y}_{t+1}$, extending the prefix or producing a completed solution. We consider a two-model setup consisting of two classes of models $\mathcal{M}$: (1) *strong expensive LLM* $\mathcal{M}_s$ that produce high-quality responses but incurs a per-token cost (2) *cheap LLM* $\mathcal{M}_w$ which offer relatively lower-quality responses at negligible cost. This abstraction captures the trade-off between generation quality and inference cost that motivates routing across heterogeneous language models.

We aim to learn a routing policy $\pi$ that, at each reasoning step $t$, chooses an action $a_t \in \{\texttt{continue}, \texttt{regenerate}\}$, to either accept the weak model's continuation or replace it with the strong model's generation. Formally, for $M_w \in \mathcal{M}_w$ and $M_s \in \mathcal{M}_s$

- If $a_t = \texttt{continue}$, the next prefix updates as $\mathbf{y}_{1:t+1} = (\mathbf{y}_{1:t}, M_w(\mathbf{y}_{1:t}))$
- If $a_t = \texttt{regenerate}$, the policy instead sets $\mathbf{y}'_{1:t} = (\mathbf{y}_{1:t-1}, M_s(\mathbf{y}_{1:t-1}))$, and then continues with $\mathbf{y}_{1:t+1} = (\mathbf{y}'_{1:t}, M_w(\mathbf{y}'_{1:t}))$.

For $\mathbf{y}_{1:t} \in C_t$, choosing $a_t = \texttt{regenerate}$ yields $\mathbf{y}'_{1:t} = (\mathbf{y}_{1:t-1}, M_s(\mathbf{y}_{1:t-1}))$, while choosing $a_t = \texttt{continue}$ leaves the prefix unchanged; in either case, the reasoning process terminates.

The quality of intermediate steps in a reasoning trace can be estimated by assigning probabilistic scores. This can be achieved through methods such as self-verification, process reward models (PRMs), or other step-level evaluation techniques. For our experiments, we adopt a PRM that evaluates partial traces and produces a sequence of step-level scores. Formally, given a reasoning trace $\mathbf{y}_{1:t}$, the PRM assigns step-level rewards as $\mathbf{r}_{1:t} = \text{PRM}(\mathbf{y}_{1:t}) = (r_1, r_2, \ldots, r_t)$, where $r_i = (\mathbf{r}_{1:t})_i$ denotes the reward for step $i$.

We use these scores as proxies for step-level correctness, providing a signal for whether escalation to a larger model is likely to offer additional benefit given the current prefix. In practice, the PRM score for a solution can also be aggregated across steps using either the product of all step scores or the minimum score across steps (Wang et al., 2023; Lightman et al., 2023). These aggregated scores are widely used in practice for ranking and comparing multiple candidate solutions. Although PRMs have been used in prior work primarily to improve candidate selection in beam search or to shape

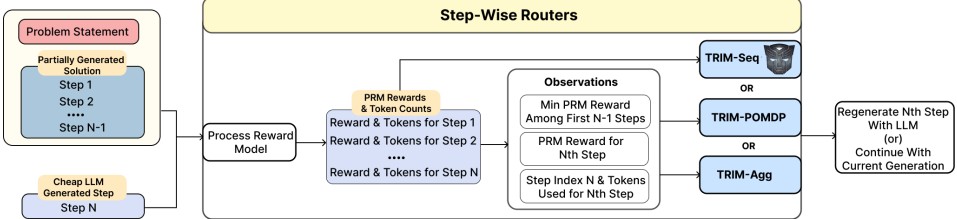

Figure 3: **Step-Wise Router architecture for TRIM** using process rewards to evaluate partial solutions and uses RL-based policies or POMDP-based solvers for making routing decisions.

exploration in search and RL (Snell et al., 2024; Setlur et al., 2024b), our use is distinct: we leverage PRM outputs to inform routing decisions during generation.

## 4 TRIM AND ROUTING STRATEGY DESIGNS

We introduce TRIM, a framework for step-level routing within each query, illustrated in Figure 1. Instead of routing an entire query to a strong model, TRIM operates at the level of individual reasoning steps. At each step $t$ of the reasoning process, the cheap model $M_w$ proposes a candidate continuation $y_t^w = M_w(\mathbf{y}_{1:t-1})$. The router policy then evaluates the partial reasoning trace together with $y_t^w$ and either accepts this step ($a_t = \texttt{continue}$) or escalates to the strong model $M_s$ ($a_t = \texttt{regenerate}$), which regenerates the step as $y_t^s = M_s(\mathbf{y}_{1:t-1})$. Based on this decision, the appended step is $y_t = y_t^w$ if $a_t = \texttt{continue}$ and $y_t = y_t^s$ otherwise. Thus, TRIM incrementally constructs the solution trace by appending at each position either the $M_w$-generated step or the $M_s$-regenerated step, depending on the router's action (see Appendix D for a detailed discussion of the action design). This contrasts with query-level routing, which which commits to either $M_w$ or $M_s$ for the entire generation.

We now describe different strategies for learning routing policies within TRIM. These strategies differ in how much information they incorporate about the reasoning trajectory and whether the decisions are myopic (local to a given step) or non-myopic (awareness of the full trajectory). On one end of the spectrum, thresholding policies rely only on current step correctness, while other approached the utilize RL training and POMDP-based formulations account for long-horizon accuracy/cost tradeoffs.

### 4.1 TRIM-THR: MYOPIC THRESHOLDING POLICY

We first introduce a simple yet effective routing policy that solely relies on the PRM score of the current generated step of the cheap model $M_w$ to make routing decisions. If this probability falls below a predefined threshold $k$, the router regenerates the step with the strong model $M_s$; otherwise, it accepts the cheap model's output and continues generation. Adjusting the threshold parameter $k$ provides a principled way to vary the task performance-cost trade-off. Our TRIM-Thr can be viewed as an adaptation of the fixed-threshold mechanism in Liao et al. (2025) to the routing setting, where thresholds vary with the cost budget. Formally, the policy is

$$\pi_{\text{thr},k}(\mathbf{y}_{1:t}) = \begin{cases} \texttt{regenerate}, & \text{if } \text{PRM}(\mathbf{y}_{1:t})_t < k, \\ \texttt{continue}, & \text{otherwise.} \end{cases} \quad (1)$$

### 4.2 RL-TRAINED POLICIES

While TRIM-Thr demonstrates strong performance (Figure 2), it is inherently *myopic* in the sense that it makes routing decisions solely based on the correctness estimate of the most recent step, without considering past context or future consequences. A richer set of signals can be exploited for more effective decision-making. For instance, even if the most recent step is predicted to be incorrect, regenerating it with a strong model may not be beneficial if the overall trajectory is already far from the correct solution, or if the additional cost of intervention outweighs the potential gain. Conversely, when prior steps are largely consistent and the trace remains plausibly aligned with a correct solution, targeted intervention can be highly impactful.

Prior work on stepwise verification (Lightman et al., 2023; Wang et al., 2023) and beam search reranking highlights the importance of leveraging the sequence of correctness scores accumulated over the reasoning trace, rather than focusing exclusively on the most recent one. Additionally, token

counts at each step provide information about the cost of regenerating with $M_s$. Incorporating these richer signals allows the router to reason jointly about (i) whether the trace remains plausibly on track toward a correct solution and (ii) whether the cost of intervention is justified, enabling more principled stepwise routing policies beyond TRIM-Thr.

**TRIM-Seq: Learning to Route from Sequential Features.** Formally, let $\mathbf{c}_{1:t} = (c_1, \ldots, c_t)$ denote the token counts for each step $y_1, \ldots, y_t$ in the reasoning trace $\mathbf{y}_{1:t}$ and $\mathbf{r}_{1:t} = (r_1, \ldots, r_t)$ be the stepwise correctness scores, then joint feature sequence is $\mathbf{f}_{1:t} = \big((r_1, c_1), \ldots, (r_t, c_t)\big)$. These features capture the two quantities that are fundamental to routing decisions in multi-step reasoning: (i) semantic fidelity, via step-level correctness estimates that indicate whether the current trajectory remains on track, and (ii) marginal intervention cost, via token counts that quantify the expense of regenerating a step with the strong model. Together, they are sufficient to represent the relevant trade-off faced by the router—whether an intervention is likely to improve final correctness enough to justify its cost—without requiring access to raw token sequences or model internals.

We parameterize the routing policy with a transformer-based network that processes the feature sequence $\mathbf{f}_{1:t}$ and outputs a distribution over actions $a_t \in \{\texttt{continue}, \texttt{regenerate}\}$. The policy is optimized via RL: each regeneration action (`regenerate`) on a prefix $\mathbf{y}_{1:t}$ incurs a cost proportional to the number of tokens generated by the strong model $M_s$, $\lambda \cdot |M_s(\mathbf{y}_{1:t-1})|$, where $\lambda > 0$ is a cost-performance trade-off parameter, and $|M_s(\mathbf{y}_{1:t-1})|$ denotes the token length of the strong model's regenerated output. We measure cost solely in terms of large-model decode tokens, as these constitute the dominant and irreducible component of inference latency and compute. While prefill (KV-cache construction) can be amortized via chunked prefilling, executed in parallel with small-model decoding–similar to the parallelization used in speculative and draft-and-verify decoding—large-model decode tokens impose unavoidable sequential cost and cannot be parallelized across tokens. The episodic return further includes the (binary) terminal task reward $R$, which reflects the correctness of the final solution. The policy is thus optimized to maximize the expected return

$$J(\pi) = \mathbb{E}_\pi \left[ R(\mathbf{y}_{1:T}) - \lambda \sum_{t=1}^{T} \mathbf{1}\{a_t = \texttt{regenerate}\} \cdot |M_s(\mathbf{y}_{1:t-1})| \right],$$

which balances solution correctness against the cumulative generation cost of invoking $M_s$.

**TRIM-Agg: Learning to Route from Aggregated Features.** TRIM-Seq leverages the full sequence of stepwise correctness scores $(r_1, r_2, \ldots, r_{t-1})$ together with token lengths to model routing decisions. While this provides rich sequential context, simpler feature representations can capture the most salient statistics of the reasoning trace. In multi-step reasoning, errors exhibit a compounding structure: a single incorrect step can invalidate the entire solution (making the minimum score a useful "weakest-link" indicator), while a sequence of mildly suspect steps can collectively cause the trajectory to drift off course (making the product of scores a useful proxy for cumulative validity across the trace). Prior work (Lightman et al., 2023; Wang et al., 2023) formalizes this intuition, showing that solution-level correctness can be effectively estimated from stepwise PRM scores via reductions such as the minimum or product. Motivated by this, we construct a reduced feature set $\tilde{\mathbf{f}}_{1:t} = \big(r_t, \min(\mathbf{r}_{1:t-1}), c_t, t\big)$, where $\min(\mathbf{r}_{1:t-1}) = \min(r_1, \ldots, r_{t-1})$, $c_t$ denotes the token length of the current step, and $t$ indexes the position in the trace. This reduced representation discards the full sequential history while retaining key aggregated indicators of correctness and cost. Using $\tilde{\mathbf{f}}_{1:t}$, we train a policy network with the same RL objective as TRIM-Seq. Empirically, this design yields substantially faster training, with negligible performance loss across all tradeoff parameters $\lambda$.

## 4.3 TRIM-POMDP: POMDP-BASED ROUTER POLICY

A final remaining challenge in stepwise routing methods discussed above arises from the imperfect nature of process reward model (PRM) estimates. While PRMs provide informative signals about the correctness of intermediate steps, their predictions are often noisy and can misclassify correct steps as incorrect (or vice versa). When trained on large amounts of data, routing policies like TRIM-Agg can implicitly learn to discard errors in the PRM estimates. However, RL under long-horizon sparse rewards makes training such policies both sample-inefficient and expensive. This raises a key question: how can routing decisions be improved when only noisy correctness signals are available?

Our solution is to explicitly treat PRM scores as imperfect observations of an unobserved latent state reflecting true correctness of the reasoning trajectory, and attempt to infer the true latent space first when learning a routing policy (akin to control in a partially-observed MDP). As shown in Figure 4,

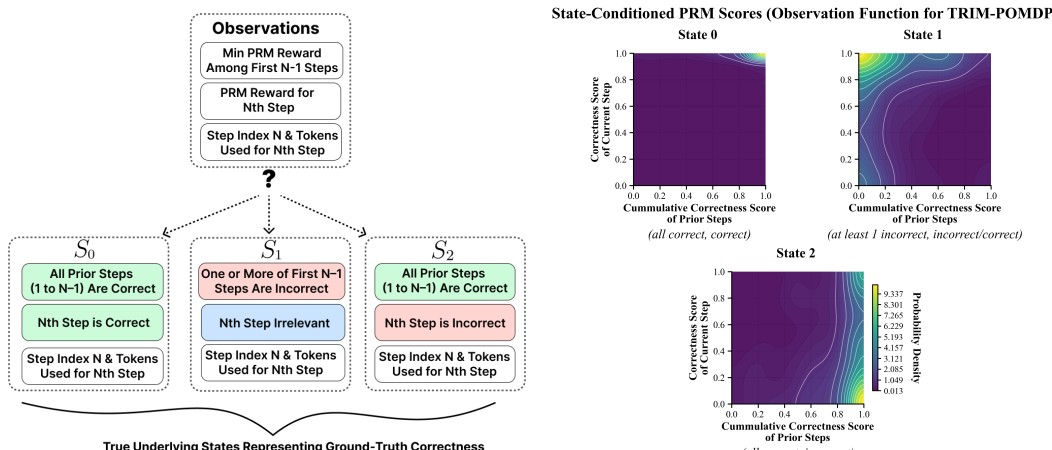

Figure 4: **POMDP state space $S$ classes and observation space $\Omega$ in TRIM-POMDP.** The latent state space $S$ consists of three correctness classes: $S_0$ (trajectory correct so far), $S_1$ (trajectory irrecoverably incorrect due to an earlier error), and $S_2$ (most recent step incorrect but prior steps correct and thus potentially recoverable), augmented with current step index and token cost. The observation space $\Omega$ comprises the PRM-based cumulative correctness score of the prior steps, PRM score of the current step, and auxiliary features, which together serve as noisy observations of the underlying latent correctness state.

Figure 5: **Learned observation function illustrating PRM noise.** Heatmaps show the empirical probability density of PRM-based observations conditioned on each latent correctness state, estimated from ProcessBench (Omni-MATH). While density concentrates around state-consistent regions (e.g., near $(1, 0)$ for $S_2$, where prior steps are correct but the current step is incorrect), the substantial spread reflects noise in PRM scores. If PRM estimates were perfectly accurate, each distribution would collapse to a single point; the observed variance motivates treating PRM outputs as noisy observations in TRIM-POMDP.

TRIM-POMDP defines the latent state using three correctness classes (augmented with the current step index and token cost): (i) $S_0$, where the trajectory remains correct so far, (ii) $S_1$, where the trajectory has already diverged irrecoverably, and (iii) $S_2$, where the most recent step is incorrect but prior steps are correct, leaving the trajectory still potentially recoverable. If this latent state were perfectly observed, the routing problem would reduce to solving a fully observable MDP. In practice, however, the latent correctness state is hidden, and we only observe noisy proxies provided by the PRM. To bridge this gap, we learn an *observation function* that helps us map the entire history of observations ($\tilde{\mathbf{f}}_{1:t}$) to a probability distribution over the latent states. Concretely, this amounts to modeling the distribution of PRM outputs conditioned on state states. Figure 5 visualizes these conditional distributions, showing that while PRM scores tend to concentrate around state-consistent regions, they exhibit substantial spread, reflecting noise in step-level correctness estimates. This observation function can be fit offline using process supervision datasets with ground-truth step-level annotations, such as ProcessBench (Zheng et al., 2024). Moreover, because the observation function only requires aligning PRM scores with annotated correctness labels, it can be trained once and reused across different performance–cost trade-off parameters $\lambda$. Once learned, we can invoke a POMDP solver on-the-fly to compute routing policies that optimally balance accuracy and cost. A detailed analysis of TRIM's robustness to PRM noise and miscalibration is provided in Appendix E.

This compact POMDP formulation of the sequential routing problem enable efficient policy computation using standard POMDP solvers. Moreover, policy computation with modern POMDP solvers is both efficient and flexible, with offline solvers typically requiring less than a minute runtime. As a result, policies can be recomputed easily for different performance–cost trade-off parameters $\lambda$. A further advantage of TRIM-POMDP is that the resulting routing policy is largely agnostic to the specific choice of LLMs ($M_s$, $M_w$), depending only on their next-step accuracies provided as inputs to the transition function. See Appendix B for further details and the complete POMDP formulation. Figure 3 illustrates the end-to-end TRIM pipeline, showing how step-level correctness signals and token costs from partially generated solutions are transformed into routing observations and used by different router instantiations (TRIM-Seq, TRIM-Agg, and TRIM-POMDP) to decide whether to regenerate a specific step with the strong model or continue with the cheap model; we next evaluate how these design choices translate into performance–cost trade-offs.

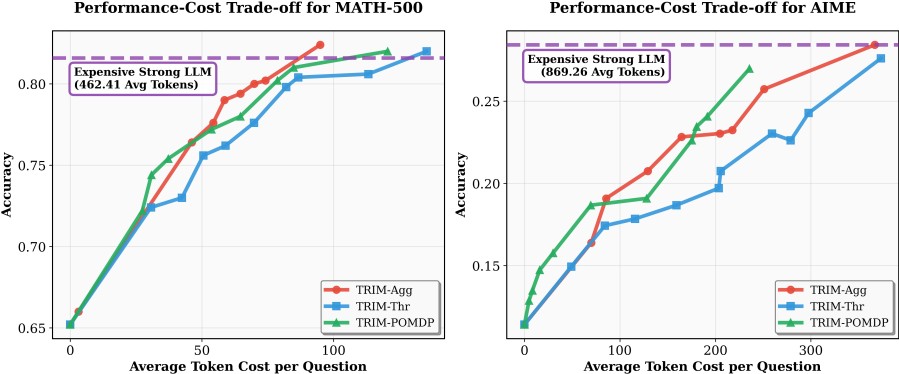

Figure 6: **Performance–cost trade-offs of TRIM routing approaches on MATH-500 and AIME.** TRIM-Thr provides a strong low-complexity baseline, while TRIM-Agg and TRIM-POMDP offer the best cost–performance trade-offs, achieving near-$M_s$ accuracy with a small fraction of expensive tokens. Notably, TRIM-POMDP performs particularly well in low-budget regimes, whereas TRIM-Agg slightly dominates at higher budgets.

## 5 EXPERIMENTS

We evaluate TRIM by benchmarking its stepwise routing strategies against established query-level routing approaches, primarily RouteLLM (Ong et al., 2024), Smoothie (Guha et al., 2024), and AutoMix (Aggarwal et al., 2023). RouteLLM is a state-of-the-art approach for query-level routing, which uses preference data for making these decisions. Smoothie instead adopts a label-free, unsupervised routing approach, fitting a latent-variable Gaussian model over output embeddings to estimate per-sample model quality and select the most suitable LLM. AutoMix, in contrast, adopts a non-predictive routing paradigm: it first utilizes smaller, cost-efficient models and then few-shot self-verification signals as observations in a POMDP policy that decides whether escalation to a larger model is necessary. We begin by introducing the evaluation metrics used throughout our analysis.

**Metrics.** We evaluate router policies by quantifying the trade-off between task performance and the cost of invoking the strong model $M_s$. In our setting, the cost of a query is measured by the number of tokens generated by the expensive (strong) model $M_s$. Importantly, TRIM does not introduce significant wall-clock overhead: when implemented using the same system-level optimizations employed by low-latency speculative decoding, TRIM is often faster than running $M_s$ alone (see Appendix A). We adapt evaluation metrics from prior work to the setting of stepwise routing in multi-step reasoning. For a query $q$ in the set of queries $\mathcal{Q}$, let $C(q; \pi)$ denote the number of tokens generated by the strong model $M_s$ under router policy $\pi$, and let $C_s(q)$ be the number of tokens generated when using $M_s$ alone. We define $\bar{C}(\pi)$ as the average number of $M_s$ tokens per query and $c(\pi)$ as the normalized fraction of tokens from $M_s$:

$$\bar{C}(\pi) = \frac{1}{|\mathcal{Q}|} \sum_{q \in \mathcal{Q}} C(q; \pi), \quad c(\pi) = \frac{\sum_{q \in \mathcal{Q}} C(q; \pi)}{\sum_{q \in \mathcal{Q}} C_s(q)}. \tag{2}$$

Following Ong et al. (2024), if $s(q; \pi) \in \{0, 1\}$ denotes the correctness of query $q$ under $\pi$, the average performance and the *performance gap recovered (*PGR*)* are defined as

$$r(\pi) = \frac{1}{|\mathcal{Q}|} \sum_{q \in \mathcal{Q}} s(q; \pi), \quad \text{PGR}(\pi) = \frac{r(\pi) - r(M_w)}{r(M_s) - r(M_w)}, \tag{3}$$

where $r(M_s)$ and $r(M_w)$ denote the accuracies of the $M_s$ and $M_w$, respectively. PGR$(\pi)$ quantifies how much of the performance gap between $M_w$ and $M_s$ is recovered by $\pi$.

To capture the cost to achieve a desired level of performance, we utilize *cost–performance threshold (*CPT*)*. Specifically, CPT$(x\%)$ denotes the minimum token cost (in terms of $\bar{C}$ or $c$) required by policy $\pi$ to achieve a PGR of $x\%$, providing a measure of efficiency at different target performance levels. Finally, following Aggarwal et al. (2023), we report the *incremental benefit per cost (IBC)* as:

$$\text{IBC}(\pi) = \frac{r(\pi) - r(M_w)}{\bar{C}(\pi)}, \quad \text{IBC}_{\text{Base}} = \frac{r(M_s) - r(M_w)}{\frac{1}{|\mathcal{Q}|} \sum_{q \in \mathcal{Q}} C_s(q; \pi)}, \quad \Delta_{\text{IBC}}(\pi) = \frac{\text{IBC}(\pi) - \text{IBC}_{\text{Base}}}{\text{IBC}_{\text{Base}}} \tag{4}$$

| Method | MATH-500 | | | | AIME | | | |
|---|---|---|---|---|---|---|---|---|
| | CPT(50%) | CPT(80%) | CPT(95%) | $\Delta_{IBC}$ | CPT(50%) | CPT(80%) | CPT(95%) | $\Delta_{IBC}$ |
| BERT | 196.60 (42.52%) | 331.45 (71.68%) | 394.49 (85.31%) | 0.08 | 331.85 (38.18%) | 616.53 (70.93%) | 701.58 (80.71%) | 0.44 |
| MF | 160.83 (34.78%) | 324.13 (70.10%) | 432.60 (93.55%) | 0.49 | 358.81 (41.28%) | 602.62 (69.32%) | 813.28 (93.56%) | 0.65 |
| SW Ranking | 185.74 (40.17%) | 279.47 (60.44%) | 330.89 (71.56%) | 0.37 | 297.08 (34.18%) | 496.77 (57.15%) | 715.72 (82.34%) | 0.79 |
| Smoothie | 220.69 (47.73%) | 345.19 (74.65%) | 433.77 (93.81%) | 0.30 | 396.75 (45.65%) | 704.50 (81.05%) | 822.09 (94.57%) | 0.03 |
| Automix | 175.67 (37.99%) | 314.11 (67.93%) | 420.87 (91.02%) | 0.12 | 450.3 (51.8%) | 715.34 (82.29%) | 857.89 (98.69%) | 0.0004 |
| AutoMix-PRM | 110.42 (23.88%) | 198.73 (42.98%) | 249.49 (53.96%) | 0.95 | 380.48 (43.77%) | 605.83 (69.7%) | 703.77 (80.96%) | 0.07 |
| TRIM-Thr | 43.68 (9.45%) | 73.74 (15.95%) | 115.99 (25.08%) | 4.75 | 204.01 (23.47%) | 314.7 (36.2%) | 372.79 (42.89%) | 1.81 |
| TRIM-Agg | 33.74 (7.3%) | **56.49 (12.22%)** | **79.58 (17.21%)** | 5.67 | **107.39 (12.35%)** | 241.55 (27.79%) | 330.42 (38.01%) | 2.50 |
| TRIM-POMDP | **29.27 (6.33%)** | 66.63 (14.41%) | 83.12 (17.98%) | **5.86** | 139.21 (16.01%) | **206.06 (23.71%)** | **244.86 (28.17%)** | **5.00** |

Table 1: Comparison of TRIM across AIME & MATH-500 benchmarks.

| Method | OlympiadBench | | | | Minerva Math | | | |
|---|---|---|---|---|---|---|---|---|
| | CPT(50%) | CPT(80%) | CPT(95%) | $\Delta_{IBC}$ | CPT(50%) | CPT(80%) | CPT(95%) | $\Delta_{IBC}$ |
| BERT | 367.75 (55.03%) | 584.14 (87.41%) | 642.50 (96.14%) | -0.04 | 209.99 (48.99%) | 378.24 (88.25%) | 421.44 (98.33%) | -0.1 |
| MF | 369.15 (55.24%) | 522.68 (78.21%) | 601.77 (90.05%) | -0.07 | 166.99 (38.96%) | 249.82 (58.28%) | 326.06 (76.07%) | 0.42 |
| SW Ranking | 351.34 (52.57%) | 511.08 (76.48%) | 635.05 (95.03%) | 0.07 | 212.73 (49.63%) | 342.71 (79.96%) | 421.17 (98.26%) | 0.04 |
| Smoothie | 348.59 (52.16%) | 511.64 (76.56%) | 615.49 (92.10%) | -0.08 | 234.66 (54.75%) | 345.16 (80.53%) | 402.19 (93.83%) | -0.09 |
| Automix | 300.4 (44.95%) | 475.37 (71.13%) | 644.46 (96.44%) | 0.02 | 85.35 (19.91%) | 193.04 (45.04%) | 231.59 (54.03%) | 0.72 |
| AutoMix-PRM | 265.95 (39.8%) | 411.47 (61.57%) | 481.49 (72.05%) | 0.22 | 72.08 (16.82%) | 140.24 (32.72%) | 196.12 (45.76%) | 1.35 |
| TRIM-Thr | 136.64 (20.45%) | 220.70 (33.03%) | 313.89 (46.97%) | 1.31 | 65.15 (15.2%) | 92.78 (21.65%) | 148.55 (34.66%) | 2.23 |
| TRIM-Agg | **94.4 (14.13%)** | **190.11 (28.45%)** | **287.17 (42.97%)** | 2.57 | **47.37 (11.05%)** | **89.54 (20.89%)** | **138.67 (32.35%)** | 3.12 |

Table 2: Cross-Benchmark Generalization of Routers Trained on AIME

Here, IBC($\pi$) measures performance improvement per unit expensive-model $M_s$ token usage, IBC$_{Base}$ is the baseline corresponding to always using $M_s$, and $\Delta_{IBC}(\pi)$ quantifies relative gain. A positive $\Delta_{IBC}$ indicates more cost-effective gains than a standalone LLM. For evaluation, we compute $\Delta_{IBC}$ across 100 equally sized performance regions between $M_w$ and $M_s$ and report the average.

**Experimental setup.** We use a two-model setup with Qwen2.5-3B-Instruct as the cheap LLM ($M_w$) and Claude 3.7 Sonnet as the expensive model ($M_s$), guided by Qwen2.5-Math-PRM-7B for step-level correctness estimation. For AIME (Di Zhang, 2025), we use an approximately 50–50 train–test split across alternate years and problem sets; for MATH (Hendrycks et al., 2021), we train on the 7.5k official training set and evaluate on MATH-500. We benchmark three TRIM routing strategies (TRIM-Thr, TRIM-Agg, and TRIM-POMDP) against AutoMix, Smoothie, and RouteLLM (Ong et al., 2024) approaches (BERT classifier, matrix factorization, and SW ranking); see Appendix G for the detailed experimental setup. To enable a fair comparison with TRIM and strengthen AutoMix, we introduce our own adapted variant of AutoMix, in which we replace the original self-verification component with cumulative PRM scores during both training and evaluation. This modification strengthens AutoMix, as PRM-based verification—designed for stepwise mathematical reasoning—yields more informative correctness signals than generic self-verification, leading AutoMix-PRM to consistently outperform vanilla AutoMix across benchmarks (Tables 1 and 2).

**Results.** Table 1 reports the evaluation results across benchmarks, and Figure 6 illustrates the performance-cost trade-off curves achieved by different TRIM strategies. To further assess the generalization capability of TRIM-Agg, we evaluate routers trained on AIME in a cross-dataset setting, testing on other math benchmarks of comparable difficulty, namely OlympiadBench and Minerva Math, and report results in Table 2 and the corresponding performance curves in Figure 7. Additional comparisons based on budgeted accuracy are provided in Appendix C.

Our experiments reveal distinct strengths across regimes. As shown in Figure 6, in the low-budget setting (large $\lambda$), TRIM-POMDP achieves superior performance via principled long-horizon planning under uncertainty. Unlike RL-trained policies, which struggle in this regime due to sparse rewards, modern POMDP solvers efficiently compute policies without being hindered by sparse-reward learning dynamics. In the high-budget regime (small $\lambda$), however, RL-trained TRIM-Agg policies performs strongly, achieving 95% of the performance gap for MATH-500 while using $\sim 80\%$ fewer expensive tokens, as policy optimization becomes significantly easier. Even our simplest approach, TRIM-Thr, achieves $5\times$ ($\Delta_{IBC} = 4.75$ vs $\Delta_{IBC} = 0.95$) better cost efficiency than baselines.

Beyond budget regimes, our cross-dataset evaluations reveal a key distinction: predictive query-level routers (e.g., RouteLLM) often fit to the intrinsic characteristics of specific datasets, while TRIM captures transferable routing behaviors that generalize across benchmarks of comparable difficulty. Query-level routers rely on coarse input-level signals and may exploit superficial correlates

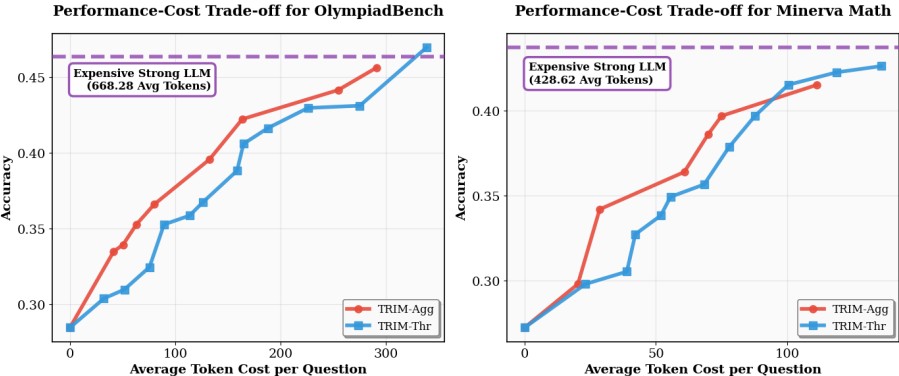

Figure 7: **Performance–cost trade-offs under dataset generalization.** TRIM-Agg routers trained on AIME demonstrate strong performance across benchmarks of similar difficulty

of difficulty (e.g., dataset style, formatting, or typical problem length) that do not transfer, whereas TRIM conditions decisions on step-level correctness signals within the evolving solution trace, which more directly reflect universal failure modes in multi-step reasoning (e.g., divergence at critical steps) and therefore generalize better across datasets. For instance, BERT achieves a $\Delta_{\text{IBC}}$ of $0.44$ on AIME but drops dramatically to $-0.04$ on OlympiadBench and $-0.1$ on Minerva Math, while SW Ranking similarly degrades from $0.79$ to $0.07$ and $0.04$, respectively. In contrast, TRIM-Agg achieves a $\Delta_{\text{IBC}}$ of $2.5$ on AIME and also maintains strong performance with $\Delta_{\text{IBC}}$ values of $2.57$ on OlympiadBench and $3.12$ on Minerva Math when trained solely on AIME, demonstrating superior generalization.

Although AutoMix (both vanilla and PRM variants) generalizes better than RouteLLM, its in-distribution performance on AIME is notably weaker; nonetheless, our PRM variant consistently outperforms vanilla AutoMix. We attribute this gap to less reliable cumulative correctness score estimates on harder problems, limiting routing accuracy despite explicit noise modeling in the correctness estimates. Overall, these findings suggest that step-level difficulty patterns captured by TRIM reflect fundamental properties of multi-step reasoning rather than dataset-specific artifacts.

Despite being trained on fewer than 500 AIME samples, TRIM-Agg achieves strong performance on the held-out test set with $38.01\%$ expensive token usage at $\text{CPT}(95\%)$ and exhibits robust generalization to datasets of comparable difficulty, consistently surpassing both TRIM-Thr and query-level baselines. In parallel, TRIM-POMDP performs strongly across all cost–performance trade-off regimes on both MATH-500 and AIME, despite its observation function being trained on different (but comparably difficult) math datasets (detailed in Appendix G) and the solver requiring only the estimated next-step accuracies of $(M_w, M_s)$ as input. Additional evidence of TRIM's robustness across different model pairs and its extension beyond math reasoning is presented in Appendix F.

## 6 DISCUSSION AND CONCLUSION

In this work, we present TRIM, an approach for targeted stepwise routing that escalates only critical steps to stronger, more expensive LLMs, intervening precisely where the partial reasoning trace risks diverging from a correct solution. Our key insight is that even a small number of well-placed interventions can dramatically boost task accuracy, enabling significant efficiency gains compared to conventional query-level routing. Building on this insight, we designed multiple routing strategies for TRIM that differ in how much trajectory-level information they exploit when making routing decisions. While TRIM already surpasses contemporary routing methods and performs competitively with oracle query-level routers, further improvements may be possible by moving beyond step-level granularity to token-level routing. Since certain tokens disproportionately influence downstream generation (Wang et al., 2025), token-level routing offers a promising direction for achieving even finer-grained and cost-efficient interventions.

## 7 REPRODUCIBILITY STATEMENT

In order to foster reproducibility of our work, we outline implementation details of our approach in Appendix G and Section 5.

## 8 ETHICS STATEMENTS

All authors have read and agree to abide by the ICLR Code of Ethics. We acknowledge the usage of LLMs for editing (e.g., grammar, spelling, word choice) purposes for polishing writing. However, the use of LLMs was limited to editing and formatting text only.

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

# Appendix Contents

## A    LATENCY ANALYSIS OF TRIM

Targeted intervention enables TRIM to reduce cost, which is defined as the number of tokens generated from the expensive, stronger LLM. In practice, TRIM can be implemented using the same system-level optimization techniques that enable low-latency speculative decoding (Leviathan et al., 2023; Liao et al., 2025; Cai et al., 2024; Pan et al., 2025), thereby avoiding frequent context re-encoding ("prefill") when switching between models and effectively ignoring the associated latency cost overhead. We now provide direct empirical evidence to this claim, demonstrating that **TRIM does not introduce significant wall-clock overhead**—and is faster than running the expensive model $M_s$ alone.

**Experimental Setup.**    We conducted end-to-end latency and throughput measurements on a fixed 2×H100 setup using vLLM with prefix caching enabled. For TRIM-Thr, GPU memory was allocated as follows: the large model $M_s$ was sharded across both GPUs using tensor parallelism (55% memory utilization per GPU), while the small model $M_w$ and the PRM each ran on separate GPUs with 40% memory utilization. For single–large-model baselines, both GPUs were fully dedicated to serving the large model. All measurements were conducted on the MATH-500 test set, and we evaluated TRIM-Thr under two model-pair configurations:

- **TRIM-Thr (1.5B + 32B):** Qwen2.5-1.5B-Instruct as $M_w$ and Qwen2.5-32B-Instruct as $M_s$
- **TRIM-Thr (1.5B + 7B):** Qwen2.5-1.5B-Instruct as $M_w$ and Qwen2.5-7B-Instruct as $M_s$

These results show that TRIM-Thr (1.5B + 32B) achieves a $1.4\times$–$2.75\times$ speedup over running the 32B model alone ($17.10/12.10 \approx 1.41$, $17.10/6.21 \approx 2.75$), despite incorporating PRM evaluation and stepwise routing. Moreover, we observe that latency gains increase with larger strong models and lower routing thresholds.

**Theoretical Justification.**    In TRIM, the small, inexpensive model $M_w$ acts as the draft generator, while the strong model $M_s$ performs chunked prefilling of the ongoing $M_w$ generation to maintain a synchronized KV cache. Consequently, routing decisions do not introduce sequential stalls: when escalation occurs, $M_s$ can immediately resume decoding from its pre-computed prefix.

Furthermore, for *all three TRIM variants* (TRIM-Thr, TRIM-Agg, and TRIM-POMDP), the routing decision overhead can be made negligible. TRIM-Agg employs a lightweight MLP with two hidden layers of 128 units, whose inference cost is effectively zero relative to LLM decoding. For TRIM-POMDP, we use SARSOP, an offline POMDP solver. To achieve the best performance out of the POMDP solver, we recompute the policy at every routing decision by initializing the initial state distribution with the current belief distribution (one can skip recomputation when belief changes are small without any loss in performance). However, because the true state space is small and each policy computation is fast (about 5 seconds on average), this enables us to precompute a lookup table mapping belief distributions to actions, resulting in zero latency from the router during inference.

With this implementation, overall wall-clock time is dominated by strong-model decoding. Because TRIM substantially reduces the number of tokens generated by $M_s$, end-to-end inference is faster than running the expensive model alone.

Table 3: End-to-end latency and throughput analysis.

| Model Configuration | Threshold ($k$) | Latency (sec/query) | Throughput (tok/sec) |
|---|---|---|---|
| **Baselines** | | | |
| Qwen2.5-32B | – | 17.10 | 31.77 |
| Qwen2.5-7B | – | 10.27 | 65.20 |
| **TRIM-Thr (1.5B + 32B)** | 0.1 | 6.21 | 64.30 |
| | 0.4 | 9.02 | 52.30 |
| | 0.7 | 12.10 | 47.86 |
| **TRIM-Thr (1.5B + 7B)** | 0.1 | 6.35 | 66.66 |
| | 0.4 | 8.17 | 65.25 |
| | 0.7 | 9.51 | 64.61 |

## B    FORMALIZING TRIM-POMDP

**Foundations of POMDPs.** A Partially Observable Markov Decision Process (POMDP) provides a principled framework for sequential decision-making under uncertainty when the underlying system state is not directly observable. A POMDP is defined by the tuple $(S, A, T, R, \Omega, \mathcal{O})$, where $S$ is the state space, $A$ the set of actions, and $\Omega$ the observation space. The transition function $T(s'|s, a)$ specifies the probability of transitioning from $s \in S$ to $s' \in S$ given $a \in A$, while the observation function $\mathcal{O}$ maps $s \in S$ and $a \in A$ to a probability distribution over the observation space $\Omega$. The reward function $R(s, a)$ assigns a scalar reward to state–action pairs.

Since the true state is hidden, the agent maintains a *belief state* $b \in \Delta(S)$, a probability distribution over latent states, updated recursively via Bayes' rule after each action–observation pair. A policy is a mapping $\pi : b \mapsto a$, and the objective is to maximize expected discounted return by optimizing over policies.

### B.1    FORMAL DEFINITION OF TRIM-POMDP

We define the components for TRIM-POMDP as follows:

**State space ($S$):** As shown in Figure 4, we categorize the state space into three correctness classes: $S_0$ (all prior steps correct and current step correct), $S_1$ (at least one prior step incorrect), and $S_2$ (prior steps correct but current step incorrect), along with a terminal absorbing state $S_{\text{ter}}$. Each state is augmented with the step index $t$ and the token count $c_t$ of the current step $\mathbf{y}_t$ within the trace $\mathbf{y}_{1:t}$.

**Observation Space ($\Omega$):** The observation space (noisy state observations) is defined as the set of aggregated features $\tilde{\mathbf{f}}_{1:t} = (r_t, \ \min(\mathbf{r}_{1:t-1}), \ c_t, \ t)$, identical to the state features used in TRIM-Agg. These observations help us obtain a probability distribution over the state space $S$.

**Action Space ($A$):** Similar to prior router policies, the action set is $A = \{\texttt{continue}, \texttt{regenerate}\}$.

**Transition Function ($T$):** The transition dynamics capture the accuracy of $M_s$ and $M_w$, as well as transitions into the terminal state $S_{\text{ter}}$. Formally:

$$\mathcal{T}(s' \in S_0 \mid s \in S_0 \cup S_2, a = \texttt{regenerate}) = p_s \quad \text{(next step accuracy of } M_s\text{)}$$
$$\mathcal{T}(s' \in S_0 \mid s \in S_0, a = \texttt{continue}) = p_w \quad \text{(next step accuracy of } M_w\text{)}$$
$$\mathcal{T}(s' \in S_1 \mid s \in S_2, a = \texttt{continue}) = 1$$
$$\mathcal{T}(s \in S_1 \mid a \in A, s' \in S_1) = 1 \quad \text{(irrecoverability assumption)}$$

**Observation Function ($\mathcal{O}$):** The observation function is a mapping from $s \in S$ to the probability distribution over the observation space $\Omega$. The probabilities $P(o|s)$, give us the likelihood of observing $o \in \Omega$ (i.e., $\tilde{\mathbf{f}}_{1:t}$) given state $s \in S$.. This corresponds to modeling the distribution of PRM scores conditioned on each state class, which can be learned from process supervision datasets with step-level annotations (e.g., PRM800K).

**Reward Function ($R$):** Invoking the strong model incurs a cost proportional to the number of tokens generated, i.e., $R(s, a = \texttt{regenerate}, s') = -\lambda \cdot |M_s(\mathbf{y}_{1:t-1})|.$. Any transition into the terminal

Table 4: **Budgeted-accuracy comparison** of TRIM across the AIME & MATH-500 benchmarks.

| Method | MATH-500 | | | | | AIME | | | | |
|---|---|---|---|---|---|---|---|---|---|---|
| | 10% | 15% | 20% | 25% | 30% | 10% | 15% | 20% | 25% | 30% |
| BERT | 66.60% (8.5%) | 67.84% (16.1%) | 69.20% (24.4%) | 70.00% (29.3%) | 70.40% (31.7%) | 13.06% (9.7%) | 14.28% (16.8%) | 15.56% (24.4%) | 17.22% (34.1%) | 18.37% (40.9%) |
| MF | 68.46% (19.9%) | 70.35% (31.4%) | 71.40% (37.8%) | 71.86% (40.6%) | 72.20% (42.7%) | 14.94% (20.7%) | 16.39% (29.3%) | 17.01% (32.9%) | 17.43% (35.4%) | 18.26% (40.2%) |
| SW Ranking | 67.82% (16.0%) | 68.20% (18.3%) | 68.80% (22.0%) | 70.01% (29.3%) | 71.14% (36.2%) | 13.90% (14.6%) | 15.10% (21.7%) | 16.18% (28.0%) | 17.43% (35.4%) | 18.88% (43.9%) |
| Smoothie | 67.20% (12.2%) | 67.77% (15.7%) | 68.20% (18.3%) | 69.35% (25.3%) | 70.00% (29.3%) | 13.07% (9.8%) | 13.90% (14.6%) | 15.15% (22.0%) | 16.18% (28.0%) | 16.60% (30.5%) |
| AutoMix-PRM | 68.88% (22.4%) | 70.54% (32.5%) | 72.19% (42.7%) | 73.78% (52.3%) | 75.26% (61.3%) | 12.74% (7.8%) | 13.85% (14.3%) | 14.95% (20.8%) | 15.88% (26.3%) | 16.80% (31.7%) |
| TRIM-Thr | 74.22% (55.0%) | 77.55% (75.3%) | 80.44% (93.0%) | 80.76% (94.8%) | 82.22% (103.8%) | 17.46% (35.6%) | 18.12% (39.4%) | 19.01% (44.7%) | 21.24% (57.8%) | 23.00% (68.1%) |
| TRIM-Agg | 76.42% (68.4%) | 79.94% (89.9%) | 82.15% (103.3%) | 84.59% (118.3%) | 87.04% (133.2%) | 19.14% (45.4%) | 20.82% (55.3%) | 22.87% (67.4%) | 23.23% (69.5%) | 25.95% (85.5%) |
| TRIM-POMDP | 76.39% (68.2%) | 78.75% (82.6%) | 81.21% (97.7%) | 81.86% (101.6%) | 82.51% (105.5%) | 18.80% (43.4%) | 19.26% (46.1%) | 22.50% (65.2%) | 25.76% (84.4%) | 28.62% (101.2%) |

Table 5: Cross-benchmark generalization performance of AIME-trained routers under budgeted-accuracy evaluation.

| Method | OlympiadBench | | | | | Minerva Math | | | | |
|---|---|---|---|---|---|---|---|---|---|---|
| | 10% | 15% | 20% | 25% | 30% | 10% | 15% | 20% | 25% | 30% |
| BERT | 29.63% (6.6%) | 30.11% (9.3%) | 30.83% (13.3%) | 31.41% (16.5%) | 32.62% (23.3%) | 28.68% (8.9%) | 29.04% (11.1%) | 29.84% (15.9%) | 31.02% (23.1%) | 31.25% (24.4%) |
| MF | 29.78% (7.4%) | 30.96% (14.0%) | 31.11% (14.9%) | 31.56% (17.4%) | 32.59% (23.1%) | 29.89% (16.2%) | 30.88% (22.2%) | 32.35% (31.1%) | 32.35% (31.1%) | 32.72% (33.3%) |
| SW Ranking | 30.95% (14.0%) | 32.44% (22.3%) | 33.04% (25.6%) | 33.63% (28.9%) | 34.07% (31.4%) | 28.31% (6.7%) | 29.41% (13.3%) | 31.40% (25.4%) | 31.25% (26.0%) | 31.51% (26.0%) |
| Smoothie | 29.63% (6.6%) | 29.93% (8.3%) | 30.86% (13.5%) | 32.30% (21.5%) | 33.33% (27.3%) | 28.31% (6.7%) | 28.68% (8.9%) | 29.15% (11.8%) | 30.15% (17.8%) | 30.51% (20.0%) |
| AutoMix-PRM | 29.76% (7.3%) | 31.34% (16.1%) | 32.61% (23.3%) | 33.83% (30.0%) | 34.75% (35.2%) | 33.66% (39.0%) | 34.86% (46.3%) | 36.48% (56.1%) | 37.88% (64.5%) | 39.54% (74.6%) |
| TRIM-Thr | 31.91% (19.3%) | 35.53% (39.5%) | 37.22% (48.9%) | 40.70% (68.4%) | 42.08% (76.0%) | 32.80% (33.8%) | 35.43% (49.7%) | 39.34% (73.3%) | 41.81% (88.3%) | 42.49% (92.4%) |
| TRIM-Agg | 35.56% (39.7%) | 37.74% (51.8%) | 39.67% (62.6%) | 42.30% (77.3%) | 43.00% (81.2%) | 35.17% (48.1%) | 37.25% (60.7%) | 40.25% (78.8%) | 41.33% (85.4%) | 42.41% (91.9%) |

state $S_{\text{ter}}$ yields the task reward $\mathbf{R}$, if and only if the final state corresponds to a correct solution (i.e., $s \in S_0$).

TRIM-POMDP can be viewed as an extension of Aggarwal et al. (2023) to the setting of multi-step reasoning. In their official implementation, Automix employs a greedy approximation to the POMDP, which is sufficient for task-level routing since it is a single-step decision problem (horizon of one). In contrast, multi-step reasoning requires planning over long horizons, making a full POMDP formulation more appropriate and motivating the use of sophisticated solvers. Furthermore, our formulation introduces key differences that provide additional flexibility. In Automix, self-verification probabilities (observations) are used to obtain estimates of model performance metrics (state features), and thus retraining of the observation function $\mathcal{O}$ is required whenever the model pair $(M_s, M_w)$ changes. However, TRIM-POMDP incorporates model accuracies into the transition function, eliminating the need for retraining the observation model when switching model pairs.

## C  BUDGETED ACCURACY

In addition to the Cost–Performance Threshold (CPT) and $\Delta_{\text{IBC}}$ metrics, we report budgeted accuracy in Table 4 and Table 5. Along with performance accuracy at each token budget, we also report the performance gap recovered (PGR) for improved comparability, with PGR percentages shown in parentheses. The columns indicate the normalized percentage of tokens generated by $M_s$, expressed relative to the total number of tokens that would be produced when running $M_s$ alone ($c(\pi)$). Budgeted accuracy is a standard metric that evaluates each routing method under a fixed compute or token budget. These results enable direct comparison under matched compute budgets and complement our CPT and IBC analyses.

One could adopt a more conservative cost model by explicitly accounting for prefill overhead. Empirically, for large-scale models, the prefill cost per token is approximately 10% to 15% of the sequential decode cost. Under a conservative 20% overhead model, the reported Cost–Performance Thresholds (CPT) for TRIM would increase by a constant factor of $1.2\times$ (or $6/5$). Importantly, this adjustment does not alter the qualitative conclusions: TRIM continues to dominate query-level baselines by a wide margin, and the relative ordering between methods remains unchanged.

## D  ANALYSIS OF regenerate ACTION DESIGN CHOICE

A central design choice in TRIM is the definition of the regenerate action: after escalating a single reasoning step to the strong model $M_s$, control is returned to the weak model $M_w$ rather than allowing $M_s$ to take over all remaining steps. We now provide both empirical motivation and ablation results supporting this choice.

Our choice to return control to $M_w$ after a single $M_s$ intervention is motivated by an empirical observation (also supported by Yu et al. (2025); Lin et al. (2024)): when an incorrect step causes the

Table 6: **Distribution of accuracy gains by number of strong-model interventions** on MATH-500 and AIME.

| Dataset | Threshold $(k)$ | $M_s$ Rounds: 1 | $M_s$ Rounds: 2 | $M_s$ Rounds: 3 | $M_s$ Rounds: $> 3$ |
|---|---|---|---|---|---|
| 3*MATH-500 | 0.10 | 19 | 8 | 2 | 1 |
| | 0.35 | 30 | 17 | 4 | 7 |
| | 0.75 | 28 | 24 | 20 | 8 |
| 3*AIME-Test | 0.10 | 14 | 9 | 5 | 0 |
| | 0.35 | 16 | 13 | 8 | 12 |
| | 0.75 | 20 | 15 | 18 | 13 |

Table 7: **Comparison of standard TRIM-Thr against One-Step Thr**, an alternative regenerate action in which $M_s$ generates all remaining steps after intervention.

| Dataset | Method | CPT(50%) | CPT(80%) | CPT(95%) | $\Delta_{\text{IBC}}$ |
|---|---|---|---|---|---|
| 2*MATH500 | TRIM-Thr | 43.68 (9.45%) | 73.74 (15.95%) | 115.99 (25.08%) | 4.75 |
| | One-Step Thr | 114.60 (24.78%) | 153.59 (33.22%) | 179.31 (38.78%) | 1.10 |
| 2*AIME | TRIM-Thr | 204.01 (23.47%) | 314.70 (36.20%) | 372.79 (42.89%) | 1.81 |
| | One-Step Thr | 217.70 (25.04%) | 482.48 (55.50%) | 590.83 (67.97%) | 0.84 |
| 2*OlympiadBench | TRIM-Thr | 136.64 (20.45%) | 220.70 (33.03%) | 313.89 (46.97%) | 1.31 |
| | One-Step Thr | 199.10 (29.79%) | 348.12 (52.09%) | 442.12 (66.16%) | 0.63 |
| 2*MinervaMath | TRIM-Thr | 65.15 (15.20%) | 92.78 (21.65%) | 148.55 (34.66%) | 2.23 |
| | One-Step Thr | 144.86 (33.80%) | 189.97 (44.32%) | 259.03 (60.43%) | 0.50 |

reasoning trajectory to diverge, correcting just that critical step is often sufficient to steer the solution back onto a successful path. In our experiments, we observed that when $M_w$ produces an incorrect step, the primary issue is not that all subsequent steps require $M_s$, but rather that the trajectory has diverged at a critical decision point. When that specific step is regenerated by $M_s$, the corrected token sequence frequently realigns the reasoning process, after which $M_w$ is able to continue successfully without additional intervention.

This behavior aligns with prior observations in multi-step reasoning (Wang et al., 2025; Bigelow et al., 2025), where a small number of pivotal steps disproportionately determine downstream correctness, and correcting these critical tokens can dramatically change the outcome. This phenomenon is empirically supported by our intervention analysis in Table 6, which lists the number of MATH-500 and AIME problems solved exclusively due to TRIM-Thr intervention (i.e., not solved by $M_w$), grouped by the number of intervention rounds, and shows that the vast majority of accuracy gains arise from just 1–3 targeted interventions rather than sustained takeover by $M_s$.

### D.1 ALTERNATIVE regenerate ACTION: FULL TAKEOVER ABLATION

We further evaluate an alternative regenerate action definition in which $M_s$ takes over generation for all remaining steps after the first escalation. We denote this variant as One-Step Thr and compare it directly with the standard TRIM-Thr used throughout the paper, reporting the results in Table 7.

This ablation shows that allowing $M_s$ to take over the remaining steps is substantially less cost-efficient, requiring significantly more expensive-model tokens to reach comparable performance. Overall, these results confirm that targeted, single-step interventions are both sufficient and optimal for correcting reasoning trajectories in multi-step reasoning tasks, validating our design choice.

### E ROBUSTNESS OF TRIM TO PRM NOISE AND MISCALIBRATION

A key strength of stepwise routing is its ability to leverage Process Reward Models (PRMs) to make fine-grained decisions during generation. TRIM is specifically designed to remain robust to the inevitable inaccuracies and miscalibration in PRM-based step-level correctness estimates, addressing this challenge through both its POMDP-based formulation and its RL-trained routing variants.

Table 8: Performance comparison of TRIM methods under noisy PRM scores.

| Method | CPT(50%) | CPT(80%) | CPT(95%) | $\Delta_{\text{IBC}}$ |
|---|---|---|---|---|
| TRIM-Thr | 187.37 (27.63%) | 450.77 (66.46%) | 564.92 (83.29%) | 0.38 |
| TRIM-Agg | 156.96 (23.14%) | 262.82 (38.75%) | 305.01 (44.97%) | 1.16 |

Table 9: **Performance of TRIM-Thr for the model pair (Qwen2.5-1.5B-Instruct, Qwen3-32B)** on AIME & MATH-500.

| | MATH-500 | | | | AIME | | | |
|---|---|---|---|---|---|---|---|---|
| Method | CPT(50%) | CPT(80%) | CPT(95%) | $\Delta_{\text{IBC}}$ | CPT(50%) | CPT(80%) | CPT(95%) | $\Delta_{\text{IBC}}$ |
| TRIM-Thr | 30.02 (4.77%) | 117.32 (18.65%) | 169.9 (27.01%) | 7.46 | 541.35 (32.61%) | 870.86 (52.47%) | 1035.62 (62.39%) | 1.25 |

In TRIM-POMDP, the routing problem is formulated as a Partially Observable Markov Decision Process (POMDP), explicitly designed to account for inaccuracies in PRM-based step-level correctness estimates. Rather than treating PRM scores as ground-truth indicators of correctness, TRIM-POMDP interprets them as noisy observations of an underlying latent correctness state. An observation function is learned to map these imperfect PRM scores to a belief distribution over these hidden correctness states, allowing routing decisions to be made based on posterior beliefs rather than raw scores. This explicit modeling of uncertainty is the primary motivation for the POMDP formulation and makes TRIM-POMDP robust to PRM miscalibration, in contrast to threshold-based routing methods such as TRIM-Thr, which operate directly on raw PRM outputs.

On the other hand, TRIM-Agg provides a complementary form of robustness and implicitly learns to model the unreliability in the PRM estimates during RL training. Thus, while the performance of TRIM-Thr can degrade using noisier PRMs, TRIM-Agg maintains robust performance. To validate this, we trained TRIM-Agg on MATH-500 using noisy PRM scores and compared its performance against TRIM-Thr on OlympiadBench, with both methods relying on the noisy PRM evaluations. Noise was introduced by applying left-padding instead of the recommended right-padding during PRM evaluation, which introduces significant noise into step-level correctness estimates.

The results in Table 8 show that TRIM-Agg maintains strong cross-dataset generalization performance despite the noisy PRM: it achieves $\text{CPT}(95\%)$ on OlympiadBench using only 45% of expensive tokens, whereas TRIM-Thr suffers significant degradation, requiring over 80% of expensive tokens to reach $\text{CPT}(95\%)$. This empirically confirms that RL-trained routing policies can also learn to be robust to PRM noise, while simple threshold-based methods cannot.

## F  GENERALIZABILITY OF TRIM ACROSS MODEL PAIRS AND DOMAINS

TRIM is model-agnostic and applies to arbitrary model pairs $(M_w, M_s)$, provided they are sufficiently compatible for stepwise continuation. We evaluate generalization along two axes: (i) robustness across model pairs, and (ii) applicability beyond mathematical reasoning.

**Generalization Across Model Pairs.** In addition to the cross-dataset generalization experiments reported in Table 2 and Figure 7, we evaluate TRIM-Thr using a substantially different model pair: Qwen2.5-1.5B-Instruct as the weak model $M_w$ and Qwen3-32B as the strong model $M_s$. Table 9 reports the resulting performance on MATH-500 and AIME.

These results demonstrate that TRIM-Thr continues to achieve strong efficiency gains under a different model family and scale, confirming that its routing behavior is not tied to a specific model pair.

**Beyond Mathematical Reasoning.** While our main experiments focus on mathematical reasoning due to the availability of reliable Process Reward Models (PRMs), TRIM itself is not math-specific. The framework only requires a mechanism for estimating step-level correctness, not a domain-specific PRM. In domains where PRMs are unavailable, TRIM can be instantiated using model-generated self-evaluations—by prompting the strong model to score intermediate steps—similar in spirit to SpecReason (Pan et al., 2025).

Table 10: **Budgeted-accuracy comparison of TRIM-Thr for the model pair (Qwen2.5-1.5B-Instruct, Qwen3-32B)** on AIME & MATH-500.

| Method | MATH-500 | | | | | AIME | | | | |
|---|---|---|---|---|---|---|---|---|---|---|
| | 10% | 15% | 20% | 25% | 30% | 10% | 15% | 20% | 25% | 30% |
| TRIM-Thr | 63.45% (70.2%) | 67.19% (75.6%) | 70.63% (80.7%) | 77.13% (90.2%) | 85.23% (102.1%) | 10.76% (25.8%) | 11.37% (27.4%) | 14.06% (34.5%) | 15.57% (38.5%) | 18.42% (46.0%) |

Table 11: **Performance of TRIM-Thr on GPQA-DIAMOND** for the model pair (Qwen3-1.7B, Qwen3-32B).

| Method | CPT(50%) | CPT(80%) | CPT(95%) | $\Delta_{\text{IBC}}$ |
|---|---|---|---|---|
| TRIM-Thr | 537.05 (28.72%) | 1012.97 (54.17%) | 1165.85 (62.34%) | 0.26 |

To demonstrate this, we extend TRIM-Thr beyond mathematics and evaluate it on GPQA-DIAMOND, a challenging benchmark spanning biology, physics, and chemistry. We use Qwen3-1.7B as $M_w$, Qwen3-32B as $M_s$, and obtain step-level correctness scores by prompting $M_s$ to assign scores on a 0–9 scale. The resulting performance–cost trade-offs are shown in Table 11. TRIM-Thr achieves CPT(95%) using only 62% of expensive-model tokens. Moreover, at a high threshold (9), TRIM-Thr improves the accuracy of Qwen3-32B from 0.419 to 0.450 while using only 73% of the expensive tokens. These results indicate that TRIM remains effective even in scientific reasoning domains without a dedicated PRM.

## G TRIM IMPLEMENTATION DETAILS

Targeted intervention enables TRIM to reduce cost, which is defined as the number of tokens generated from the expensive, stronger LLM. While this sort of "step-level" routing may appear to require frequent context re-encoding ("prefill") when switching models, this overhead can be largely amortized by chunked prefilling of the large model's KV cache in parallel with the small model's decoding and PRM evaluation. In practice, the large model can maintain a synchronized KV cache (aggregate prefill) by incrementally prefilling the tokens generated by the small model. This ensures that upon escalation, the large model can almost immediately transition to decoding the next token rather than re-encoding the history. This design mirrors speculative decoding strategies (Leviathan et al., 2023; Cai et al., 2024), which similarly overlap draft and verification phases to minimize latency.

In contrast, the generation (decode) phase of the large model imposes an unavoidable sequential cost and cannot be parallelized across tokens, while the prefill can be parallelized, cached, or reused across steps. Consequently, the number of tokens generated by the large model is the dominant and irreducible component of inference cost, making it both a theoretically clean and operationally grounded efficiency metric. In summary, large-model decode tokens capture the true marginal expense, while prefilling and routing overheads are effectively hidden by overlapping computation with small-model decoding and PRM evaluation. While we did not implement these system-level optimizations in our primary experiments—as public API endpoints for proprietary models (e.g., Claude) do not currently expose the necessary KV cache state management—these techniques are established standards in open-source serving frameworks (e.g., vLLM, SGLang) and are widely utilized by production providers. Consequently, the absence of these optimizations in our API-based evaluation does not undermine the conceptual validity of our cost model.

**TRIM-Agg Implementation.** The router policy is parameterized by a simple MLP policy with two hidden layers (128 units each, Tanh activations), followed by separate actor and critic heads, and trained using PPO. The selected hyperparameters are summarized in Table 12. Training is conducted across performance–cost trade-off parameters $\lambda$, ranging from $3 \times 10^{-4}$ to $8 \times 10^{-5}$ for AIME and from $8 \times 10^{-4}$ to $3 \times 10^{-4}$ for MATH, at regular intervals.

**TRIM-POMDP Implementation.** For TRIM-POMDP, we learn the observation function using a reflected KDE estimator applied to the ProcessBench (Zheng et al., 2024) dataset. Specifically, we evaluate our PRM on step-by-step solutions in the dataset and align its outputs with human-annotated step-level labels. The observation function is trained on Omni-MATH problems, while evaluation is conducted on AIME and GSM8k for MATH-500. Importantly, the only model-specific information required by TRIM-POMDP is the next-step accuracies of the models, which are estimated from the

Table 12: **Hyperparameters** used for TRIM-Agg.

| Hyperparameter | Value |
|---|---|
| Learning rate | 1.0e-4 |
| Clipping coefficient | 0.2 |
| Entropy coefficient | 0.01 |
| Advantages | Unnormalized |
| Rewards | Undiscounted |
| GAE | 0.95 |

corresponding training sets. For solving the POMDP, we use SARSOP (Kurniawati et al., 2008), which we implement using the `POMDPs.jl` framework (Egorov et al., 2017). We solve the POMDP using SARSOP (Kurniawati et al., 2008), implemented using the `POMDPs.jl` framework (Egorov et al., 2017), with hyperparameters set to the default values provided in `POMDPs.jl`. While SARSOP can in principle handle large observation spaces, it is sensitive to the choice of the initial state distribution. To achieve the best performance, we therefore recompute the policy at every step using the updated belief distribution as the initial state distribution. To improve efficiency, we employ a simple heuristic: the policy is recomputed only if the belief mass on state $S_2$ (the case where the most recent step is incorrect but prior steps are correct) lies within 0.35–0.40 of the maximum belief state class; otherwise, we default to continuing with $M_w$ (i.e., $a_t = \texttt{continue}$). For all TRIM routing policies, the reasoning trace is truncated to a maximum of 30 steps during both training and inference, and the solution at this cutoff is returned.

**AutoMix Implementation.** In its official implementation, AutoMix employs a greedy approximation to the proposed POMDP formulation. For a two-model setup, queries are categorized into three classes: (1) solvable by $M_w$, (2) unsolvable by both $M_w$ and $M_s$, and (3) solvable only by $M_s$. Using few-shot self-verification or correctness scores, AutoMix discretizes the verifier probability space into uniformly spaced bins. For each bin, it estimates the empirical conditional distribution over these three outcome classes–essentially learning a binned classifier that maps verifier probabilities to class likelihoods. During inference, routing decisions are made greedily: given a new verifier score, the corresponding bin provides the estimated probability distribution over outcome classes, and the router selects an action that maximizes the expected return under a specified trade-off parameter balancing accuracy and cost.

**Smoothie Implementation.** For Smoothie, we include results for Smoothie-Train, following the official implementation. As noted in its limitations (Guha et al., 2024), Smoothie is optimized for performance rather than explicit cost–performance trade-offs and is designed to escalate to the strongest available model within a given mixture of LLMs. To populate a performance-cost curve in our setting, we therefore route based on the estimated difference in LLM quality scores for each sample. Since Smoothie requires outputs from at least three models to estimate these scores during training, we use Mathstral-7B-v0.1 as the third model in our Smoothie experiments.

## H EFFECTIVENESS OF QWEN2.5-MATH-PRM-7B FOR AIME

A natural question is how effective Qwen2.5-Math-PRM-7B is as a PRM for AIME problems, which are substantially more challenging than other mathematical benchmarks considered in this work, such as MATH-500, OlympiadBench, and Minerva Math. Since step-level ground-truth annotations are not available for the AIME, we assess PRM effectiveness by measuring how well aggregated stepwise PRM scores correlate with the final correctness of full solutions. Our evaluation procedure is as follows:

1. We generate full reasoning traces for AIME, MATH-500, and OlympiadBench using the same weak model, Qwen2.5-3B-Instruct.

2. For each trace, we compute PRM step-level scores and aggregate them using the minimum score across steps. This aggregation strategy is used throughout our work and is standard in prior PRM literature.

3. We compute the AUC-ROC between these aggregated scores and the final correctness labels, quantifying how well PRM scores separate correct from incorrect solutions.

The resulting AUC-ROC values are:

- AIME: 0.8397
- OlympiadBench: 0.8954
- MATH500: 0.9363

These scores indicate that while PRM reliability is strongest on MATH-500 and OlympiadBench for the generated traces, it still provides meaningful discriminatory signal on AIME–far above random (0.5) and sufficiently informative for decision-making. The somewhat lower AUC on AIME reflects that these problems are more difficult and produce longer, noisier reasoning traces, but the PRM remains useful rather than failing outright.

Crucially, the reduced PRM reliability on AIME highlights the need for robustness to noisy correctness estimates. TRIM-POMDP is explicitly designed for this: by modeling PRM scores as noisy observations of an underlying correctness state, it remains stable even when the PRM is imperfect. Consistent with this robustness role, TRIM-POMDP achieves a $2.76\times$ improvement over TRIM-Thr in $\Delta_{\text{IBC}}$ on AIME—much larger than the $1.23\times$ improvement on MATH-500–demonstrating that uncertainty–aware routing is especially valuable on harder datasets where PRM noise is more pronounced, and estimates are less reliable.

