# OpenReview forum: "TRIM: Hybrid Inference via Targeted Stepwise Routing in Multi-Step Reasoning Tasks"
_ICLR.cc/2026/Conference — ICLR 2026 Poster_

### Official Review · Reviewer_oN83 · 2025-10-28

**Soundness:** 2
**Presentation:** 3
**Contribution:** 2
**Rating:** 2
**Confidence:** 4

**Summary:**

This paper proposes TRIM, a hybrid inference method that routes reasoning steps between a small and a large LLM. The key idea is to escalate only critical reasoning steps to the stronger model, improving efficiency while maintaining accuracy. The paper includes several routing strategies, from a simple threshold-based approach to more advanced RL and POMDP-based ones and demonstrates improvement over query level baseline results on multiple math reasoning benchmarks.

**Strengths:**

The overall presentation is good, and the paper is mostly easy to read and understand. The concept of step-level routing appears practical, bridging the gap between coarse query-level routing and fine-grained token-level control. The experimental setup seems reasonable, and the reward modeling and reinforcement learning training are well-motivated.

**Weaknesses:**

Some parts of the paper are confusing or underspecified, especially the step definition and regeneration process. The cost metric ignores potential latency from frequent model switching, which could matter in deployment. A few experimental details (like baseline performance numbers) seem inconsistent or not clearly justified. Clarifying these points would make the technical contribution much easier to trust and replicate. See questions section for more details. I am open to increase my rating if the issues raised are adequately addressed.

**Questions:**

1- Section 4 feels ambiguous. While Figure 3 suggests that routing and decisions happen step-by-step, the notation y_s = Ms(y1:t−1) makes it look like the strong model regenerates the whole prefix. Which interpretation is correct? The notation and explanation should be clarified.

2- What exactly is a “step”? Is it defined as a sentence, a reasoning line, or a fixed number of tokens? How do you prompt or constrain the model to generate only one step at a time? Is this step size consistent across different datasets or tasks?

3- In TRIM-Seq, how is the step-level correctness score computed? Is it purely from the PRM output or based on an external reference?

4- Since cost is measured only by the number of tokens from the expensive model, does this ignore switching overhead between models? Routing back and forth should add latency and communication cost, which could be nontrivial in real systems.

5- The reported baseline performance for Claude 3.7 seems off, while this model reportedly has ~96.2 % accuracy on MATH-500,  Figure 6 shows it closer to 80%. Can you clarify this discrepancy or explain what experimental setup causes this gap?

---

> ### Author Response · Authors · 2025-11-22
> **Rebuttal by Authors - I**
>
> Thank you for your feedback! To address your concerns, we (i) clarify that routing and decisions occur step-by-step and formally justify our definition of a "step", (ii) explain how step-level correctness score are obtained and used in TRIM-Agg, (iii) empirically demonstrate and theoretically explain that TRIM achieves lower end-to-end latency than running the expensive large model $M_s$ alone, and (iv) verify that our reported baseline performance for Claude 3.7 Sonnet is consistent with the official numbers, as we do not enable extended-thinking mode in our evaluations.
>
> > Section 4 feels ambiguous. While Figure 3 suggests that routing and decisions happen step-by-step, the notation y_s = Ms(y1:t−1) makes it look like the strong model regenerates the whole prefix. Which interpretation is correct? The notation and explanation should be clarified.
>
> Apologies for any confusion. In our formulation, each model $M$ is defined as a function that takes as input the partial reasoning trace $y_{1:t-1}$ and outputs the next step $y_t$ (Lines 174-177). The routing and decisions occur step-by-step, and the notation $y^s_t = M_s(y_{1:t-1})$ refers to the strong model $M_s$ generating the next step conditioned on the current prefix, not regenerating the entire prefix.
>
> We have now explicitly mentioned in the revised version of the paper (Line 179) that $M$ is viewed as a function in our setup.
>
> > What exactly is a “step”? Is it defined as a sentence, a reasoning line, or a fixed number of tokens? How do you prompt or constrain the model to generate only one step at a time? Is this step size consistent across different datasets or tasks?
>
> In our work, a step is defined as a segment of the model’s output delimited by a double newline ("\n\n") (Lines 158-159). This convention follows standard practice in mathematical-reasoning datasets and prior process-supervision work, where double line breaks are used to separate solution steps. In mathematical reasoning tasks, double line breaks are commonly used to segment solution steps [1,2]. We use standard CoT prompting, which naturally produces solutions in this step-segmented format. During generation, we simply pause decoding whenever the model emits "\n\n", and treat the preceding text as one step. This definition of a step is used consistently across all datasets and experiments in the paper.
>
> > In TRIM-Seq, how is the step-level correctness score computed? Is it purely from the PRM output or based on an external reference?
>
> In TRIM-Seq, the step-level correctness estimates are obtained directly from the PRM output. The PRM provides us with the individual correctness scores for each of the steps in the partially generated solution.
>
> [1] Lightman, Hunter, et al. "Let's verify step by step." The Twelfth International Conference on Learning Representations. 2023.
>
> [2] Zhang, Zhenru, et al. "The lessons of developing process reward models in mathematical reasoning." arXiv preprint arXiv:2501.07301 (2025).

---

> ### Author Response · Authors · 2025-11-22
> **Rebuttal by Authors - II**
>
> > Since cost is measured only by the number of tokens from the expensive model, does this ignore switching overhead between models? Routing back and forth should add latency and communication cost, which could be nontrivial in real systems.
>
> Thanks for pointing out the possible issues regarding latency. In practice, TRIM can be implemented using the same system optimization techniques that enable low-latency speculative decoding.  We now provide direct empirical evidence demonstrating that *TRIM does not introduce significant wall-clock overhead*—and, is faster than running the expensive model $M_s$ alone.
>
> **Experimental Setup:** We conducted end-to-end latency and throughput measurements on a fixed 2×H100 setup using vLLM with prefix caching enabled. For TRIM-Thr, GPU memory was allocated as follows: the large model $M_s$ used both GPUs with tensor parallelism (0.55 memory utilization per GPU), while the small model $M_w$ and the PRM each ran on separate GPUs (0.4 utilization each). For the single large model baselines, both GPUs were fully dedicated to serving the large model. All measurements were conducted on the MATH-500 test dataset and we evaluated TRIM-Thr with two model pair configurations:
> * TRIM-Thr (1.5B + 32B): Qwen2.5-1.5B-Instruct as $M_w$ and Qwen2.5-32B-Instruct as $M_s$
> * TRIM-Thr (1.5B + 7B): Qwen2.5-1.5B-Instruct as $M_w$ and Qwen2.5-7B-Instruct as $M_s$
>
>
> | Model Configuration | Threshold (k) | Latency (sec/query) | Throughput (tok/sec) |
> |---------------|------|-------------------|-------------------|
> | **Baselines** | | | |
> | Qwen2.5-32B | — | 17.10 | 31.77 |
> | Qwen2.5-7B | — | 10.27 | 65.20 |
> | | | | |
> | **TRIM-Thr (1.5B + 32B)** | | | |
> | | 0.1 | 6.21 | 64.30 |
> | | 0.4 | 9.02 | 52.30 |
> | | 0.7 | 12.10 | 47.86 |
> | | | | |
> | **TRIM-Thr (1.5B + 7B)** | | | |
> | | 0.1 | 6.35 | 66.66 |
> | | 0.4 | 8.17 | 65.25 |
> | | 0.7 | 9.51 | 64.61 |
>
> These results demonstrate that TRIM-Thr (1.5B + 32B) is 1.4×–2.75× faster (17.10 / 12.10 $\approx$1.41, 17.10 / 6.21 $\approx$2.75) than running the large 32B model alone, even though TRIM involves PRM evaluation and stepwise routing. Furthermore, we observe that the latency improvements for TRIM increase as the model size grows and the threshold decreases.
>
>
> **Theoretical Justification:** The small, cheap LLM $M_w$ serves as the draft model, while the strong model maintains a shadow prefill (chunked KV-cache) of the ongoing $M_w$ generation. As a result, routing decisions do not introduce sequential stalls: when escalation occurs, the strong model can immediately continue decoding from its cached prefix.
>
> Furthermore, in **all three TRIM approaches (TRIM-Thr, TRIM-Agg, TRIM-POMDP), the routing decision computation time can be made negligible**.
> * TRIM-Agg uses a small MLP with two hidden layers (128 units each), yielding effectively zero inference overhead relative to LLM decoding time.
> * For TRIM-POMDP, we use SARSOP, an offline POMDP solver. To achieve the best performance out of the POMDP solver, we recompute the policy at every routing decision by initializing the initial state distribution with the current belief distribution (one can skip recomputation when belief changes are small without any loss in performance). However, because the true state space is small and each policy computation is fast (~5 seconds), this enables us to **precompute a lookup table mapping belief distributions to actions**, resulting in zero latency from the router during inference. Instead of this, we can also apply additional heuristics to avoid policy recomputation at every step, which work well in practice and retain optimal performance (Line 739-742).
>
> With such an implementation, because large-model decode tokens dominate wall-clock time and TRIM substantially reduces the number of these tokens, the overall inference becomes faster than running the expensive model alone.
>
> > The reported baseline performance for Claude 3.7 seems off, while this model reportedly has ~96.2 % accuracy on MATH-500, Figure 6 shows it closer to 80%. Can you clarify this discrepancy or explain what experimental setup causes this gap?
>
> During generation with Claude 3.7, we do not use extended thinking, and therefore we obtain an accuracy of 82% on MATH-500, which is consistent with the value reported on the official Claude 3.7 Sonnet webpage (https://www.anthropic.com/news/claude-3-7-sonnet) for no extended thinking (82.2%).

---

> > ### Comment · Reviewer_oN83 · 2025-11-23
> >
> > Thank you to the authors for the detailed response and the new experiments. The revised manuscript is much clearer and reads well; accordingly, I am increasing my score to 4.
> >
> > However, I still have concerns about the usability and impact of the proposed techniques. Many of the methodological details appear specific to verifiable math datasets, where a reliable process reward model can be trained. For more general problem types—such as planning, coding, or tool use—the paper does not provide experimental evidence demonstrating that the approach extends beyond math. While math-solving performance is an important indicator of improved general reasoning capabilities, gains in this domain do not necessarily translate to better performance on more complex or less easily verifiable tasks.
> >
> > If improved math performance is indeed the primary goal of the paper, then the method should be compared against other state-of-the-art math reasoning approaches, not only against the base model, including baselines trained on extended reasoning datasets.

---

> ### Author Response · Authors · 2025-11-28
>
> Thanks for the comment! We would like to clarify that although our main experiments focus on mathematical reasoning due to the availability of reliable PRMs, **TRIM itself is not math-specific**: it only requires a mechanism for estimating step-level correctness, not a domain-specific PRM. **In domains where PRMs do not yet exist, TRIM can be instantiated using model-generated self-evaluations**—similar to the approach used in SpecReason [1]—by prompting the strong model to produce a score for each reasoning step.
>
>
> To demonstrate this, we extend TRIM-Thr beyond mathematics and evaluate it on GPQA-DIAMOND, a challenging benchmark spanning biology, physics, and chemistry. We use Qwen3-1.7B as $M_w$, Qwen3-32B as $M_s$, and obtain step-level scores by prompting $M_s$ to assign scores from 0 to 9. The results below show that TRIM-Thr remains effective even without a PRM:
>
>
> | Method | CPT(50%) | CPT(80%) | CPT(95%) | $Δ_{IBC}$ |
> |---------|----------|----------|----------|-------|
> | TRIM-Thr | 537.05 (28.72%) | 1012.97 (54.17%) | 1165.85 (62.34%) | 0.26 |
>
> We observe that TRIM-Thr achieves CPT(95%) using only 62% of expensive tokens. Furthermore, we also observed that TRIM-Thr improves the accuracy of Qwen3-32B from 0.419 to 0.45 while using only 73% of its expensive tokens (threshold = 9), demonstrating that TRIM remains beneficial even in scientific reasoning domains.
>
>
> [1] Pan, Rui, et al. "Specreason: Fast and accurate inference-time compute via speculative reasoning." arXiv preprint arXiv:2504.07891 (2025).

---

### Official Review · Reviewer_5mDK · 2025-10-29

**Soundness:** 2
**Presentation:** 3
**Contribution:** 2
**Rating:** 4
**Confidence:** 4

**Summary:**

The paper presents TRIM, a hybrid inference framework that uses a process reward model (PRM) to decide when to route reasoning steps from a smaller model to a larger one. The approach aims to balance inference cost and reasoning accuracy. It is primarily evaluated on mathematical reasoning tasks such as AIME using Qwen2.5-Math-PRM-7B and Claude 3.7 Sonnet. The authors introduce a Cost–Performance Threshold (CPT) metric and report improved cost–performance trade-offs compared with several baseline methods.

**Strengths:**

* **Clear motivation and relevance:** The paper tackles a practical problem of reducing inference cost while maintaining reasoning accuracy.
* **Reasonably structured framework**: The framework is generally well organized, though its design largely follows existing PRM-based routing paradigms.
* **Clarity:** The paper is well organized and easy to follow, with clear figures and explanations.

**Weaknesses:**

* **Limited novelty over existing PRM-based routing methods:** The paper does not clearly articulate how TRIM differs conceptually from prior stepwise routing approaches using process reward models (PRMs).
* **Unclear reliability of PRM supervision for AIME problems:** The use of Qwen2.5-Math-PRM-7B raises concerns about label reliability on AIME, where reasoning traces are substantially longer and harder to evaluate.
* **Model family mismatch and API issues:** Mixing models from different families (e.g., Qwen and Claude) may introduce inconsistencies in formatting and reasoning styles. In addition, Claude 3.7 Sonnet’s chat-completion API makes partial-trace continuation non-trivial.
* **Insufficient baseline coverage:** Several key comparisons are missing, such as non-routing anchors (Always-Small, Always-Large), query-level routing, and adaptive compute methods.
* **Code and data unavailable for review:** The absence of released code or data prevents verification and reproducibility of the reported results.

**Questions:**

* There is already a considerable body of work on stepwise routing with PRMs [1–3]. What is the fundamental difference between TRIM and these existing approaches?
* How effective is Qwen2.5-Math-PRM-7B as a PRM for AIME problems? Compared with the datasets reported in the official documentation [4], AIME problems are substantially more difficult, and the reasoning traces generated by LLMs are usually much longer. It is therefore more challenging to evaluate the correctness of intermediate steps.
* The Cost–Performance Threshold (CPT) metric is not commonly used. In contrast, budgeted accuracy, which compares methods under the same compute or token budget, is a standard and decision-useful metric for evaluating efficient reasoning or routing. Could the authors also report results under this metric for better comparability?
* The baseline coverage should be expanded. In addition to stepwise routing methods, please include non-routing anchors such as Always-Small-Model and Always-Large-Model, as well as query-level routing [5,6] and step-/token-level adaptive compute approaches [1–3].
* Line 420: The paper uses models from different families as the base (e.g., Qwen vs. Claude), which may have distinct formatting styles and output distributions. How do the authors handle these inconsistencies? Moreover, since Claude 3.7 Sonnet relies on a chat-completion API schema, continuing reasoning from a partial trace can differ from a true completion mode. Could the authors explain how this issue is addressed or justify this model choice?
* Figures 6 and 7: Should the x-axis represent the number of tokens from the expensive model?
* How generalizable is the proposed approach? Can the authors test TRIM with different combinations of cheap and expensive models, and on additional domains such as scientific reasoning or code to demonstrate broader applicability?
* Can the authors provide the code and data for review?

## Reference:
[1] Liao, Baohao, et al. "Reward-Guided Speculative Decoding for Efficient LLM Reasoning." Forty-second International Conference on Machine Learning.

[2] Pan, Rui, et al. "Specreason: Fast and accurate inference-time compute via speculative reasoning." arXiv preprint arXiv:2504.07891 (2025).

[3] Liu, Yuliang, et al. "AdaptiveStep: Automatically Dividing Reasoning Step through Model Confidence." Forty-second International Conference on Machine Learning.

[4] Zhang, Zhenru, et al. "The lessons of developing process reward models in mathematical reasoning." arXiv preprint arXiv:2501.07301 (2025).

[5] Ding, Dujian, et al. "Hybrid LLM: Cost-Efficient and Quality-Aware Query Routing." The Twelfth International Conference on Learning Representations.

[6] Guha, Neel, et al. "Smoothie: Label free language model routing." Advances in Neural Information Processing Systems 37 (2024): 127645-127672.

---

> ### Author Response · Authors · 2025-11-22
> **Rebuttal by Authors - I**
>
> Thank you for your feedback! To address your concerns, we (i) clarify how TRIM's objective and framework fundamentally differ from prior work [1, 2, 3], (ii) provide empirical evidence that our PRM is reasonably effective on AIME, (iii) report budgeted accuracy evaluations for all routing approaches, (iv) expand the baseline coverage, and (v) explain why TRIM remains robust despite model-family mismatch while also showing its generalizability.
>
> > What is the fundamental difference between TRIM and approaches in [1–3]?
>
> Thanks for the comment! We were previously unaware of some of these works, and we have now incorporated a corresponding discussion into the revised version of the paper (Lines 139-146).
>
> **Different Objective:** While the objective of Reward-Guided Speculative Decoding [1] and SpecReason [2] is on reducing latency and marginally improving the accuracy of the strong-expensive mode, TRIM aims to maximize accuracy under an explicit cost-performance tradeoff, where only a limited budget of strong-model tokens is allowed. Thus, TRIM is designed explicitly as a budget-aware routing framework, whereas approaches in [1,2] operate in a fixed high-budget regime and are not presented as routing methods. Therefore, SpecReason [2] and  Reward-Guided Speculative Decoding [1] are benchmarked for performance and latency against baselines such as Best-of-N, Speculative Decoding, and Always-Large-Model rather than benchmarking them against the routing approaches.
>
> **Different Framework & Setting:** [1,2] accept the small model step generations ("speculated reasoning step") only using fixed high correctness thresholds ([1] uses PRM scores while [2] prompts the expensive model to generate the score), while TRIM is a general framework that allows for decisions based on diverse policies (e.g., POMDP-based, RL-trained) optimized for any given arbitrary budget. Our TRIM-Thr can be viewed as an adaptation of the fixed-threshold mechanism in [1] to the routing setting, where thresholds vary with the cost budget.
>
> **Handling Inaccuracies in PRM scores:**  Even if [1,2] were adapted to a routing setting similar to TRIM-Thr (Lines 250-257), they–unlike TRIM-POMDP, TRIM-Agg, and TRIM-Seq–do not account for inaccuracies or poor calibration in the correctness scores and remain inherently myopic. As a result, they are not optimized for long-horizon planning  required for efficient utilization of the budget utilization in routing problems.
>
> [3] is fundamentally different from the TRIM routing framework: it focuses on training PRMs by automatically segmenting model outputs into variable-length reasoning steps based on token-level confidence, thereby improving PRM quality. It does not perform routing or make compute–allocation decisions. Thus, while complementary to our work, AdaptiveStep cannot serve as a routing baseline in our setting.
>
> > How effective is Qwen2.5-Math-PRM-7B as a PRM for AIME problems?
>
> We obtain labels for AIME from the public dataset https://huggingface.co/datasets/gneubig/aime-1983-2024.
>
> Since step-level ground-truth annotations are not available for the AIME Dataset, we assess the effectiveness of Qwen2.5-Math-PRM-7B on AIME by measuring how well its aggregated stepwise scores correlate with the final correctness of full solutions. Our evaluation procedure is as follows:
>
> 1. We generate full reasoning traces for AIME, MATH-500, and OlympiadBench using the same weak model (Qwen2.5-3B-Instruct).
> 2. For each trace, we obtain PRM step-level scores and aggregate them using the minimum score across steps, which is also used in our work and is a standard metric used in prior PRM work.
> 3. We compute the AUC-ROC between these aggregated scores and the final correctness labels, quantifying how well PRM scores separate correct from incorrect solutions.
>
> The resulting AUC-ROC values are:
> * AIME: 0.8397
> * OlympiadBench: 0.8954
> * MATH500: 0.9363
>
> These scores indicate that while PRM reliability is strongest on MATH-500 and OlympiadBench for the generated traces, it still **provides meaningful discriminatory signal on AIME**—far above random (0.5) and sufficiently informative for decision-making. The somewhat lower AUC on AIME reflects that these problems are more difficult and produce longer, noisier reasoning traces, but the PRM remains useful rather than failing outright.
>
> Crucially, the reduced PRM reliability on AIME highlights the need for robustness to noisy correctness estimates. TRIM-POMDP is explicitly designed for this: by modeling PRM scores as noisy observations of an underlying correctness state, it remains stable even when the PRM is imperfect. Consistent with this robustness role, TRIM-POMDP achieves a 2.76× improvement over TRIM-Thr in $\Delta_{IBC}$ on AIME—much larger than the 1.23× improvement on MATH-500—demonstrating that uncertainty-aware routing is especially valuable on harder datasets where PRM noise is more pronounced and estimates are less reliable.

---

> ### Author Response · Authors · 2025-11-22
> **Rebuttal by Authors - II**
>
> > Could the authors also report results under budgeted accuracy for better comparability?
>
> Thanks for the suggestion! We have now added budgeted accuracy evaluations for all routing approaches in the revised version of the paper (see Appendix C). Along with the accuracies at each token budget, we also report the performance-gap recovered (PGR) for improved comparability, with PGR percentage values provided in brackets. The columns indicate the normalized percentage of tokens generated by $M_s$, expressed relative to the total number of tokens that would be produced when running $M_s$ alone. For convenience, we additionally provide the complete set of budgeted-accuracy results below.
>
> |  | | MATH-500 | | | | | AIME | | | |
> |--------|------|------|------|------|------|------|------|------|------|------|
> | Method | 10% | 15% | 20% | 25% | 30% | 10% | 15% | 20% | 25% | 30% |
> | BERT | 66.60% (8.5%) | 67.84% (16.1%) | 69.20% (24.4%) | 70.00% (29.3%) | 70.40% (31.7%) | 13.06% (9.7%) | 14.28% (16.8%) | 15.56% (24.4%) | 17.22% (34.1%) | 18.37% (40.9%) |
> | MF | 68.46% (19.9%) | 70.35% (31.4%) | 71.40% (37.8%) | 71.86% (40.6%) | 72.20% (42.7%) | 14.94% (20.7%) | 16.39% (29.3%) | 17.01% (32.9%) | 17.43% (35.4%) | 18.26% (40.2%) |
> | SW Ranking | 67.82% (16.0%) | 68.20% (18.3%) | 68.80% (22.0%) | 70.01% (29.3%) | 71.14% (36.2%) | 13.90% (14.6%) | 15.10% (21.7%) | 16.18% (28.0%) | 17.43% (35.4%) | 18.88% (43.9%) |
> | Smoothie | 67.20% (12.2%) | 67.77% (15.7%) | 68.20% (18.3%) | 69.35% (25.3%) | 70.00% (29.3%) | 13.07% (9.8%) | 13.90% (14.6%) | 15.15% (22.0%) | 16.18% (28.0%) | 16.60% (30.5%) |
> | AutoMix-PRM | 68.88% (22.4%) | 70.54% (32.5%) | 72.19% (42.7%) | 73.78% (52.3%) | 75.26% (61.3%) | 12.74% (7.8%) | 13.85% (14.3%) | 14.95% (20.8%) | 15.88% (26.3%) | 16.80% (31.7%) |
> | TRIM-Thr | 74.22% (55.0%) | 77.55% (75.3%) | 80.44% (93.0%) | 80.76% (94.8%) | 82.22% (103.8%) | 17.46% (35.6%) | 18.12% (39.4%) | 19.01% (44.7%) | 21.24% (57.8%) | 23.00% (68.1%) |
> | TRIM-Agg | 76.42% (68.4%) | 79.94% (89.9%) | 82.15% (103.3%) | 84.59% (118.3%) | 87.04% (133.2%) | 19.14% (45.4%) | 20.82% (55.3%) | 22.87% (67.4%) | 23.23% (69.5%) | 25.95% (85.5%) |
> | TRIM-POMDP | 76.39% (68.2%) | 78.75% (82.6%) | 81.21% (97.7%) | 81.86% (101.6%) | 82.51% (105.5%) | 18.80% (43.4%) | 19.26% (46.1%) | 22.50% (65.2%) | 25.76% (84.4%) | 28.62% (101.2%) |
>
> Table: Budgeted-accuracy comparison of TRIM across the AIME & MATH-500 benchmarks.
>
> |  | | OlympiadBench | | | | | Minerva Math | | | |
> |--------|------|------|------|------|------|------|------|------|------|------|
> | Method | 10% | 15% | 20% | 25% | 30% | 10% | 15% | 20% | 25% | 30% |
> | BERT | 29.63% (6.6%) | 30.11% (9.3%) | 30.83% (13.3%) | 31.41% (16.5%) | 32.62% (23.3%) | 28.68% (8.9%) | 29.04% (11.1%) | 29.84% (15.9%) | 31.02% (23.1%) | 31.25% (24.4%) |
> | MF | 29.78% (7.4%) | 30.96% (14.0%) | 31.11% (14.9%) | 31.56% (17.4%) | 32.59% (23.1%) | 29.89% (16.2%) | 30.88% (22.2%) | 32.35% (31.1%) | 32.35% (31.1%) | 32.72% (33.3%) |
> | SW Ranking | 30.95% (14.0%) | 32.44% (22.3%) | 33.04% (25.6%) | 33.63% (28.9%) | 34.07% (31.4%) | 28.31% (6.7%) | 29.41% (13.3%) | 31.40% (25.4%) | 31.25% (24.4%) | 31.51% (26.0%) |
> | Smoothie | 29.63% (6.6%) | 29.93% (8.3%) | 30.86% (13.5%) | 32.30% (21.5%) | 33.33% (27.3%) | 28.31% (6.7%) | 28.68% (8.9%) | 29.15% (11.8%) | 30.15% (17.8%) | 30.51% (20.0%) |
> | AutoMix-PRM | 29.76% (7.3%) | 31.34% (16.1%) | 32.61% (23.3%) | 33.83% (30.0%) | 34.75% (35.2%) | 33.66% (39.0%) | 34.86% (46.3%) | 36.48% (56.1%) | 37.88% (64.5%) | 39.54% (74.6%) |
> | TRIM-Thr | 31.91% (19.3%) | 35.53% (39.5%) | 37.22% (48.9%) | 40.70% (68.4%) | 42.08% (76.0%) | 32.80% (33.8%) | 35.43% (49.7%) | 39.34% (73.3%) | 41.81% (88.3%) | 42.49% (92.4%) |
> | TRIM-Agg | 35.56% (39.7%) | 37.74% (51.8%) | 39.67% (62.6%) | 42.30% (77.3%) | 43.00% (81.2%) | 35.17% (48.1%) | 37.25% (60.7%) | 40.25% (78.8%) | 41.33% (85.4%) | 42.41% (91.9%) |
>
> Table: Cross-benchmark generalization performance of AIME-trained routers under budgeted-accuracy evaluation.

---

> ### Author Response · Authors · 2025-11-22
> **Rebuttal by Authors - III**
>
> > The baseline coverage should be expanded. In addition to stepwise routing methods, please include non-routing anchors such as Always-Small-Model and Always-Large-Model, as well as query-level routing [5,6] and step-/token-level adaptive compute approaches [1–3].
>
> In the performance–cost curves (Figures 6 and 7), Always-Small and Always-Large already appear as fixed points: Always-Small corresponds to the leftmost point on the curve, while Always-Large is shown as the horizontal dashed line. A randomized query-level router that routes the full query to the large model with probability p (approximately) traces the linear interpolation between these two anchors as p varies from 0 to 1. Such a randomized router achieves CPT(x%) $\approx$ x% and $\Delta_{IBC}\approx 0$, therefore our **$ \Delta_{IBC}$ metric directly measures improvement over the linear interpolation between the two anchors**.
>
> We have now **included the evaluation results of SMOOTHIE-TRAIN [6]** across all benchmarks in the revised version of the paper. As noted in the limitations of Smoothie (Section 6 of [6]), the method is optimized purely for performance and does not incorporate any cost-performance trade-off; the method given a mixture of models, is designed to escalate to the stronger model. To populate a performance-cost curve in our setting, we therefore route based on the estimated difference in LLM quality scores for each sample. Because Smoothie relies on unsupervised quality estimation—i.e., it predicts model quality on a sample without access to ground-truth correctness—it is expected to perform poorly on mathematical reasoning tasks, where unsupervised confidence scores are unreliable. We confirm this empirically in our experiments. (To estimate quality scores on training samples, Smoothie requires outputs from at least three models; we use Mathstral-7B-v0.1 as the third model.)
>
>
> HybridLLM [6] uses a BERT-based router trained with BART scores on the instruction following dataset MixInstruct. When adapted to our setting of mathematical reasoning tasks–replacing the BART quality metric with correctness of the final answer as the quality of model response–**this becomes exactly the BERT baseline** included in Tables 1 and 2.
>
>
> Furthermore, adaptation of SpecReason [2] to the our routing setup does not provide any meaningful change over TRIM-Thr with the only difference being in the scoring strategy for steps, which is obtained by prompting the expensive model to generate a utility score rather than generating it via a PRM. As mentioned above, [3] is quite distinct from our routing framework and instead presents an approach for training PRMs, and TRIM-Thr already corresponds to the natural adaptation of [1] to our routing context by varying cutoff thresholds.
>
>
> To further expand the baseline coverage, we also have **included the performance of Automix [7]**, an adaptive and non-predictive routing approach which first utilizes cost-efficient models and then employs a few-shot self-verification mechanism where a meta-reviewer decides whether escalation to a larger model is necessary based on the estimated reliability of the output. To enable a fair comparison with TRIM and obtain stronger baseline results, we replace the original self-verification component in AutoMix with our cumulative PRM score when evaluating its performance in the experiments.
>
> [1] Liao, Baohao, et al. "Reward-Guided Speculative Decoding for Efficient LLM Reasoning." Forty-second International Conference on Machine Learning.
>
> [2] Pan, Rui, et al. "Specreason: Fast and accurate inference-time compute via speculative reasoning." arXiv preprint arXiv:2504.07891 (2025).
>
> [3] Liu, Yuliang, et al. "AdaptiveStep: Automatically Dividing Reasoning Step through Model Confidence." Forty-second International Conference on Machine Learning.
>
> [4] Zhang, Zhenru, et al. "The lessons of developing process reward models in mathematical reasoning." arXiv preprint arXiv:2501.07301 (2025).
>
> [5] Ding, Dujian, et al. "Hybrid LLM: Cost-Efficient and Quality-Aware Query Routing." The Twelfth International Conference on Learning Representations.
>
> [6] Guha, Neel, et al. "Smoothie: Label free language model routing." Advances in Neural Information Processing Systems 37 (2024): 127645-127672.
>
> [7] Aggarwal, Pranjal, et al. "AutoMix: Automatically mixing language models." Advances in Neural Information Processing Systems 37 (2024): 131000-131034.

---

> ### Author Response · Authors · 2025-11-22
> **Rebuttal by Authors - IV**
>
> > The paper uses models from different families as the base (e.g., Qwen vs. Claude), which may have distinct formatting styles and output distributions. How do the authors handle these inconsistencies? Moreover, since Claude 3.7 Sonnet relies on a chat-completion API schema, continuing reasoning from a partial trace can differ from a true completion mode. Could the authors explain how this issue is addressed or justify this model choice?
>
> Thanks for bringing up this point! Qwen2.5-3B-Instruct and Claude 3.7 Sonnet both reliably follow Chain-of-Thought prompting conventions, and empirical inspection shows that their step delimiters ("\n\n") and reasoning formats are sufficiently compatible for stepwise continuation. Since TRIM operates at the level of semantic steps rather than strict token-level alignment, minor stylistic differences do not seem to affect performance.
>
> We chose Qwen2.5-3B and Claude 3.7 Sonnet to demonstrate the broad applicability of TRIM across heterogeneous model families, rather than relying on two tightly matched models from the same vendor. TRIM applies equally well to any model pair that supports conditional continuation from each other's partial responses.
>
> Anthropic's API allows the messages array to end with an assistant role. Although Claude 3.7 Sonnet uses a chat-completion API, it supports prefix-conditioned continuation by placing the accumulated trace directly in the assistant turn without the need for appending an instruction mentioning to continue the reasoning exactly from the given prefix. When the API sees the last message is from the assistant, it understands the task is not to respond to a user, but to continue the assistant's last turn which is the partially generated trace in our case. This is equivalent to standard prefix continuation because the chat schema does not alter the model’s ability to condition on the supplied text. We verified that the model reliably produces the next step from the provided partial solution without rewriting or reinterpreting earlier steps.
>
> >Figures 6 and 7: Should the x-axis represent the number of tokens from the expensive model?
>
> Yes, Figures 6 and 7 use the x-axis to show the average number of tokens from expensive model per question, excluding any tokens from the cheap model.
>
> >How generalizable is the proposed approach? Can the authors test TRIM with different combinations of cheap and expensive models, and on additional domains such as scientific reasoning or code to demonstrate broader applicability?
>
> The TRIM approach is model agnostic and generalizable to any model pairs as long as they are sufficiently compatible for stepwise continuation. We have demonstrated cross-dataset generalization experiments in Table 2 and Figure 7.
>
> Below, we present the TRIM-Thr results obtained using Qwen2.5-1.5B-Instruct as the cheap LLM ($M_w$) and Qwen3-32B as the expensive model ($M_s$) on MATH500 and AIME.
>
> | | MATH-500 | | | | AIME | | |
> |------|------|------|------|------|------|------|------|
> | CPT(50%) | CPT(80%) | CPT(95%) | $\Delta_{IBC}$ | CPT(50%) | CPT(80%) | CPT(95%) | $\Delta_{IBC}$ |
> | 30.02 (4.77%) | 117.32 (18.65%) | 169.9 (27.01%) | 7.46 | 541.35 (32.61%) | 870.86 (52.47%) | 1035.62 (62.39%) | 1.25 |
>
> Table: Performance of TRIM-Thr on the AIME \& MATH-500 benchmarks for the model pair (Qwen2.5-1.5B-Instruct, Qwen3-32B).
>
> | | MATH-500 | | | | | AIME | | | |
> |------|------|------|------|------|------|------|------|------|------|
> | 10% | 15% | 20% | 25% | 30% | 10% | 15% | 20% | 25% | 30% |
> | 63.45% (70.2%) | 67.19% (75.6%) | 70.63% (80.7%) | 77.13% (90.2%) | 85.23% (102.1%) | 10.76% (25.8%) | 11.37% (27.4%) | 14.06% (34.5%) | 15.57% (38.5%) | 18.42% (46.0%) |
>
> Table: Budgeted-accuracy of TRIM-Thr on the AIME & MATH-500 benchmarks for the model pair (Qwen2.5-1.5B-Instruct, Qwen3-32B).
>
> The results show that TRIM-Thr maintains strong performance for the model pair (Qwen2.5-1.5B-Instruct, Qwen3-32B) as well: it achieves CPT(95%) on MATH-500 and AIME using only 27% and 63% of expensive tokens, respectively. Furthermore, it also attains high values of $\Delta_{IBC}$, particularly on MATH-500 ($\Delta_{IBC} = 7.46$).
>
> Applying TRIM beyond math is a natural next step. There now exist early Process Reward Models for code (e.g., CodePRM), and we plan to experiment with TRIM in that setting. However, the reliability of these code PRMs is still unclear, so we are first assessing whether they can provide sufficiently stable supervision for our setting.
>
> > Can the authors provide the code and data for review?
>
> Our code is currently undergoing a legal review required prior to release. We will try to make the code and data available as soon as this process is complete.

---

> > ### Author Response · Authors · 2025-11-28
> > **Did our Rebuttal Address your Concerns?**
> >
> > Dear Reviewer,
> >
> > Given that there are only four more days left in the rebuttal period, we wanted to reach out to see if our rebuttal above addressed all your concerns or if any others still remained? We are happy to discuss further and provide more evidence in the few days that are left. If all your concerns are addressed, we would be really grateful if you could acknowledge that soon.
> >
> > Furthermore, to also demonstrate the broader applicability of TRIM across different domains, we extend TRIM-Thr beyond mathematics and evaluate it on GPQA-DIAMOND, a challenging benchmark spanning biology, physics, and chemistry. We use Qwen3-1.7B as $M_w$, Qwen3-32B as $M_s$, and obtain step-level scores by prompting $M_s$ to assign scores from 0 to 9. We would like to clarify that although our main experiments focus on mathematical reasoning due to the availability of reasonably reliable PRMs, TRIM itself is not math-specific: it only requires a mechanism for estimating step-level correctness, not a domain-specific PRM. In domains where PRMs do not yet exist, TRIM can be instantiated using model-generated self-evaluations—similar to the approach used in SpecReason [2], which was suggested by the reviewer—by prompting the strong model to produce a score for each reasoning step.
> >
> > The results below show that TRIM-Thr remains effective even without a PRM:
> >
> >
> > | Method | CPT(50%) | CPT(80%) | CPT(95%) | $Δ_{IBC}$ |
> > |---------|----------|----------|----------|-------|
> > | TRIM-Thr | 537.05 (28.72%) | 1012.97 (54.17%) | 1165.85 (62.34%) | 0.26 |
> >
> > We observe that TRIM-Thr achieves CPT(95%) using only 62% of expensive tokens. Furthermore, we also observe that TRIM-Thr improves the accuracy of Qwen3-32B from 0.419 to 0.45 while using only 73% of its expensive tokens (threshold = 9), demonstrating that TRIM remains beneficial even in scientific reasoning domains.
> >
> >
> >
> > Thanks,
> >
> > Authors

---

### Official Review · Reviewer_hcjw · 2025-11-02

**Soundness:** 3
**Presentation:** 3
**Contribution:** 3
**Rating:** 6
**Confidence:** 3

**Summary:**

This paper proposes a novel method that improves the inference efficiency of large language models by performing routing decisions at the step level. The authors formalize this process as a sequential decision problem and design several routing strategies, including a simple threshold-based policy (TRIM-Thr), two reinforcement learning–based policies (TRIM-Seq and TRIM-Agg) that balance accuracy and cost, and a POMDP-based policy (TRIM-POMDP) to handle uncertainty in step correctness estimation. Experiments on multiple mathematical reasoning benchmarks demonstrate that this approach can achieve performance comparable to strong models while significantly reducing computational cost.

**Strengths:**

1. The paper is clearly structured and well written, with intuitive figures and tables that effectively illustrate the key ideas and experimental results.
2. The proposed method demonstrates strong innovation by moving beyond traditional query-level routing, offering a substantial conceptual advance in improving multi-step reasoning efficiency, and showing significant empirical effectiveness.
3. The authors formalize the routing process as a sequential decision problem and propose multiple complementary strategies that balance methodological simplicity with theoretical depth.

**Weaknesses:**

1. The approach heavily relies on the Process Reward Model; inaccuracies or poor calibration in PRM estimation may negatively affect routing stability and overall performance.
2. Experiments focus primarily on mathematical reasoning tasks, and the generalization of TRIM to other multi-step reasoning domains (e.g., code generation or scientific reasoning) remains untested.
3. Stepwise generation introduces additional inference latency; although token cost is reduced, the paper does not provide a detailed analysis of runtime efficiency or latency trade-offs.

**Questions:**

See weakness.

---

> ### Author Response · Authors · 2025-11-22
> **Rebuttal by Authors - I**
>
> Thank you for your feedback! To address your concerns, we show that TRIM-POMDP & TRIM-Agg remain robust under noisy and inaccurate PRM estimates, and we empirically demonstrate and theoretically explain that TRIM achieves lower end-to-end latency than running the expensive large model $M_s$ alone.
>
> > The approach heavily relies on the Process Reward Model; inaccuracies or poor calibration in PRM estimation may negatively affect routing stability and overall performance.
>
> Thanks for the question! Please note that modeling the routing problem as a Partially Observable MDP precisely allows TRIM-POMDP to explicitly account for inaccuracies in PRM-based step-level correctness estimates (Lines 79-81). Accounting for noise or inaccuracies in the PRM is the main motivation for using a POMDP formulation. In TRIM-POMDP, the PRM scores are treated as noisy observations rather than ground-truth indicators of correctness, and an observation function is learned to map these imperfect scores to a distribution over these hidden correctness states. This design makes the router robust to PRM miscalibration, since decisions are made based on state-belief distributions rather than raw PRM outputs, unlike TRIM-Thr (Lines 340–352).
>
> On the other hand, TRIM-Agg implicitly learns to model the uncertainty in the PRM estimates during RL training (Line 321-322). Thus, while the performance of TRIM-Thr can degrade using noisier PRMs, TRIM-Agg maintains robust performance. To validate this, we trained TRIM-Agg on MATH-500 using noisy PRM scores and compared its performance against TRIM-Thr on OlympiadBench, with both methods relying on the noisy PRM evaluations. Noise was introduced by applying left-padding instead of the recommended right-padding during PRM evaluation, which introduces significant noise into step-level correctness estimates.
>
> | Method | CPT(50%) | CPT(80%) | CPT(95%) | $Δ_{IBC}$ |
> |---------|----------|----------|----------|-------|
> | TRIM-Thr | 187.37 (27.63%) | 450.77 (66.46%) | 564.92 (83.29%) | 0.38 |
> | TRIM-Agg | 156.96 (23.14%) | 262.82 (38.75%) | 305.01 (44.97%) | 1.16 |
>
> The results show that TRIM-Agg maintains strong cross-dataset generalization performance despite the noisy PRM: it achieves CPT(95%) on OlympiadBench using only 45% of expensive tokens, whereas TRIM-Thr suffers significant degradation, requiring over 80% of expensive tokens to reach CPT(95%). This empirically confirms that RL-trained routing policies can also learn to be robust to PRM noise, while simple threshold-based methods cannot.
>
> > Experiments focus primarily on mathematical reasoning tasks, and the generalization of TRIM to other multi-step reasoning domains (e.g., code generation or scientific reasoning) remains untested.
>
> Thanks for your feedback, that’s a good point! Applying TRIM beyond math is a natural next step. There now exist early Process Reward Models for code (e.g., CodePRM), and we plan to experiment with TRIM in that setting. However, the reliability of code PRMs is still unclear, so we are first assessing whether they can provide sufficiently stable supervision for our setting.

---

> ### Author Response · Authors · 2025-11-22
> **Rebuttal by Authors - II**
>
> > Stepwise generation introduces additional inference latency; although token cost is reduced, the paper does not provide a detailed analysis of runtime efficiency or latency trade-offs.
>
> Thanks for pointing out the possible issues regarding latency. In practice, TRIM can be implemented using the same system optimization techniques that enable low-latency speculative decoding.  We now provide direct empirical evidence demonstrating that *TRIM does not introduce significant wall-clock overhead*—and, is faster than running the expensive model $M_s$ alone.
>
> **Experimental Setup:** We conducted end-to-end latency and throughput measurements on a fixed 2×H100 setup using vLLM with prefix caching enabled. For TRIM-Thr, GPU memory was allocated as follows: the large model $M_s$ used both GPUs with tensor parallelism (0.55 memory utilization per GPU), while the small model $M_w$ and the PRM each ran on separate GPUs (0.4 utilization each). For the single large model baselines, both GPUs were fully dedicated to serving the large model. All measurements were conducted on the MATH-500 test dataset and we evaluated TRIM-Thr with two model pair configurations:
> * TRIM-Thr (1.5B + 32B): Qwen2.5-1.5B-Instruct as $M_w$ and Qwen2.5-32B-Instruct as $M_s$
> * TRIM-Thr (1.5B + 7B): Qwen2.5-1.5B-Instruct as $M_w$ and Qwen2.5-7B-Instruct as $M_s$
>
>
> | Model Configuration | Threshold (k) | Latency (sec/query) | Throughput (tok/sec) |
> |---------------|------|-------------------|-------------------|
> | Baselines | | | |
> | Qwen2.5-32B | — | 17.10 | 31.77 |
> | Qwen2.5-7B | — | 10.27 | 65.20 |
> | | | | |
> | TRIM-Thr (1.5B + 32B) | | | |
> | | 0.1 | 6.21 | 64.30 |
> | | 0.4 | 9.02 | 52.30 |
> | | 0.7 | 12.10 | 47.86 |
> | | | | |
> | TRIM-Thr (1.5B + 7B) | | | |
> | | 0.1 | 6.35 | 66.66 |
> | | 0.4 | 8.17 | 65.25 |
> | | 0.7 | 9.51 | 64.61 |
>
> These results demonstrate that TRIM-Thr (1.5B + 32B) is 1.4×–2.75× faster (17.10 / 12.10 $\approx$1.41, 17.10 / 6.21 $\approx$2.75) than running the large 32B model alone, even though TRIM involves PRM evaluation and stepwise routing. Furthermore, we observe that the latency improvements for TRIM increase as the model size grows and the threshold decreases.
>
>
> **Theoretical Justification:** The small, cheap LLM $M_w$ serves as the draft model, while the strong model maintains a shadow prefill (chunked KV-cache) of the ongoing $M_w$ generation. As a result, routing decisions do not introduce sequential stalls: when escalation occurs, the strong model can immediately continue decoding from its cached prefix.
>
> Furthermore, in **all three TRIM approaches (TRIM-Thr, TRIM-Agg, TRIM-POMDP), the routing decision computation time can be made negligible**.
> * TRIM-Agg uses a small MLP with two hidden layers (128 units each), yielding effectively zero inference overhead relative to LLM decoding time.
> * For TRIM-POMDP, we use SARSOP, an offline POMDP solver. To achieve the best performance out of the POMDP solver, we recompute the policy at every routing decision by initializing the initial state distribution with the current belief distribution (one can skip recomputation when belief changes are small without any loss in performance). However, because the true state space is small and each policy computation is fast (~5 seconds), this enables us to **precompute a lookup table mapping belief distributions to actions**, resulting in zero latency from the router during inference. Instead of this, we can also apply additional heuristics to avoid policy recomputation at every step, which work well in practice and retain optimal performance (Line 739-742).
>
> With such an implementation, because large-model decode tokens dominate wall-clock time and TRIM substantially reduces the number of these tokens, the overall inference becomes faster than running the expensive model alone.

---

> > ### Author Response · Authors · 2025-11-28
> > **Did our Rebuttal Address your Concerns?**
> >
> > Dear Reviewer,
> >
> > Given that there are only four more days left in the rebuttal period, we wanted to reach out to see if our rebuttal addressed all your concerns or if any others still remained? We are happy to discuss further and provide more evidence in the few days that are left. If all your concerns are addressed, we would be really grateful if you could acknowledge that soon.
> >
> >
> > Furthermore, to also address your concern regarding the generalization of TRIM to other multi-step reasoning domains, we would like to clarify that although our main experiments focus on mathematical reasoning due to the availability of reasonably reliable PRMs, TRIM itself is not math-specific: it only requires a mechanism for estimating step-level correctness, not a domain-specific PRM. In domains where PRMs do not yet exist, TRIM can be instantiated using model-generated self-evaluations—similar to the approach used in SpecReason [1]—by prompting the strong model to produce a score for each reasoning step.
> >
> >
> > To demonstrate this, we extend TRIM-Thr beyond mathematics and evaluate it on GPQA-DIAMOND, a challenging benchmark spanning biology, physics, and chemistry. We use Qwen3-1.7B as $M_w$, Qwen3-32B as $M_s$, and obtain step-level scores by prompting $M_s$ to assign scores from 0 to 9. The results below show that TRIM-Thr remains effective even without a PRM:
> >
> >
> > | Method | CPT(50%) | CPT(80%) | CPT(95%) | $Δ_{IBC}$ |
> > |---------|----------|----------|----------|-------|
> > | TRIM-Thr | 537.05 (28.72%) | 1012.97 (54.17%) | 1165.85 (62.34%) | 0.26 |
> >
> > We observe that TRIM-Thr achieves CPT(95%) using only 62% of expensive tokens. Furthermore, we also observe that TRIM-Thr improves the accuracy of Qwen3-32B from 0.419 to 0.45 while using only 73% of its expensive tokens (threshold = 9), demonstrating that TRIM remains beneficial even in scientific reasoning domains.
> >
> >
> > [1] Pan, Rui, et al. "Specreason: Fast and accurate inference-time compute via speculative reasoning." arXiv preprint arXiv:2504.07891 (2025).
> >
> >
> > Thanks,
> >
> > Authors

---

### Official Review · Reviewer_X7GU · 2025-11-06

**Soundness:** 3
**Presentation:** 3
**Contribution:** 3
**Rating:** 6
**Confidence:** 4

**Summary:**

This paper proposes TRIM (Targeted Routing in Multi-step reasoning tasks), a novel hybrid inference strategy designed to improve the cost-efficiency of multi-step reasoning. The central problem it addresses is that conventional "query-level" routing (assigning an entire task to either a cheap or an expensive LLM) is inefficient. Relying only on cheap models leads to cascading errors, while relying only on expensive models is cost-prohibitive.

TRIM's core idea is to perform **step-wise routing**. A cheap model generates each reasoning step by default. After each step, a router, guided by a Process Reward Model (PRM), decides whether to accept the cheap step or to regenerate *only that specific step* with an expensive model before handing control back to the cheap model. This "targeted intervention" aims to prevent cascading failures at critical steps while saving costs on routine ones.

The authors propose and evaluate four distinct routing strategies:
1.  **TRIM-Thr:** A simple threshold policy (regenerate if PRM score < *k*).
2.  **TRIM-Seq:** An RL-trained policy (using full step history).
3.  **TRIM-Agg:** A simpler RL-trained policy (using aggregated step statistics).
4.  **TRIM-POMDP:** A sophisticated policy that models the true step correctness as a hidden state and the PRM score as a noisy observation.

On mathematical reasoning benchmarks, the simple TRIM-Thr policy is shown to be 6.51x more cost-efficient than contemporary query-level routing. The more advanced RL and POMDP policies can match the performance of the expensive-only model while reducing the number of expensive tokens generated by 80%.

**Strengths:**

* **Novelty:** The core concept of step-wise routing, as opposed to query-level routing, is a significant and novel contribution. It directly targets the "cascading failure" weakness of multi-step reasoning.
* **Impressive Results:** The efficiency gains are substantial. Achieving an 80% reduction in expensive token usage while matching the strong model's accuracy is a highly compelling result. The 6.51x cost-efficiency improvement from the *simplest* policy (TRIM-Thr) over SOTA query-level routing is also very strong.
* **Methodological Thoroughness:** The paper explores four different router implementations, from a simple heuristic to a complex POMDP. This shows a deep understanding of the problem, from a practical, easy-to-implement solution to a more theoretically-grounded one that handles observational uncertainty (PRM noise).
* **Generalizability:** The cross-dataset generalization experiments are a key strength, validating that the model is learning fundamental properties of reasoning difficulty rather than just overfitting to a specific dataset's quirks.

**Weaknesses:**

* **Incomplete Cost Model (Latency):** The paper's cost model is defined *only* as "the number of tokens generated by the expensive model." This model completely ignores the very significant latency and compute costs incurred at *every single step* of the generation. The loop is: (1) $M_w$ generates a step, (2) generation pauses, (3) the 7B-parameter PRM runs, (4) the router policy runs (for TRIM-POMDP, this involves *re-solving* the policy), (5) $M_s$ *maybe* runs. This sequential, stop-and-go process introduces a massive amount of latency per step, which is not accounted for. A simple, streamed generation from the expensive model $M_s$ might be much faster in wall-clock time, even if it uses more "expensive tokens."
* **The "Regenerate" Action Definition:** The "regenerate" action is defined as $M_s$ regenerating the current step, after which control is *immediately* returned to the cheap $M_w$ for the *next* step. This seems suboptimal. If a step was so critical that $M_w$ failed and $M_s$ had to intervene, why would you trust $M_w$ with the very next step? This could lead to an immediate re-failure. A more intuitive action would be to let $M_s$ take over for the remainder of the generation. This design choice feels overly optimized for the paper's cost metric, not for maximum accuracy.
* **Complexity vs. Performance:** The TRIM-POMDP approach is by far the most complex, requiring ground-truth step-level annotations (from ProcessBench) to learn the observation function and a POMDP solver. However, in the results (e.g., Figure 6), its performance-cost curve is almost identical to that of the much simpler `TRIM-Agg` policy (a simple MLP trained with PPO on aggregated stats). The paper does not adequately justify why this massive increase in complexity is worthwhile for such a marginal gain.
* **Dependency on the PRM:** The entire system is critically dependent on the quality of the Qwen2.5-Math-PRM-7B model. While the POMDP policy *models* this uncertainty, the simpler and highly effective TRIM-Thr and TRIM-Agg policies *trust* the PRM scores. The paper doesn't analyze how the performance of these simpler policies degrades if a less-accurate, noisier, or miscalibrated PRM is used.

**Questions:**

1.  **Latency Cost:** Your cost model ignores the significant per-step latency of running the PRM and the router policy. How does the total wall-clock time of a TRIM-generated solution (with its N sequential stop-and-think steps) compare to a single, uninterrupted generation from the expensive model $M_s$?
2.  **"Regenerate" Action:** Why did you choose to return control to the weak model $M_w$ *immediately* after an expensive $M_s$ intervention? Have you experimented with an alternative "regenerate" action where $M_s$ takes over for all *remaining* steps, and how does that compare on the performance-cost curve?
3.  **PRM Sensitivity:** How sensitive are the simpler (and more practical) TRIM-Thr and TRIM-Agg policies to the quality of the Process Reward Model? How much does their performance drop if a smaller, faster, but less accurate PRM is used?
4.  **Justifying POMDP Complexity:** The results for TRIM-Agg and TRIM-POMDP in Figure 6 appear nearly identical. Given the extreme complexity of the POMDP (requiring annotated data to train the observation function, plus an online solver), what is the practical justification for using it over the much simpler RL-based TRIM-Agg policy?

---

> ### Author Response · Authors · 2025-11-22
> **Rebuttal by Authors - I**
>
> Thank you for your feedback! To address your concerns, we (i) empirically demonstrate and theoretically explain that TRIM achieves lower end-to-end latency than running the expensive large model $M_s$ alone, (ii) empirically justify our choice of the $\mathrm{regenerate}$ action definition, (iii) show that TRIM-Agg remains robust under noisy and inaccurate PRM estimates, (iv) clarify the practical motivation for TRIM-POMDP relative to TRIM-Agg.
>
>
> ## Re: Latency Comparison of TRIM and Large Expensive $M_s$ Model
>
> > How does the total wall-clock time of a TRIM-generated solution compare to the generation from the expensive model $M_s$?
>
> Thanks for pointing out the possible issues regarding latency. In practice, TRIM can be implemented using the same system optimization techniques that enable low-latency speculative decoding.  We now provide direct empirical evidence demonstrating that *TRIM does not introduce significant wall-clock overhead*—and is faster than running the expensive model $M_s$ alone.
>
> **Experimental Setup:** We conducted end-to-end latency and throughput measurements on a fixed 2×H100 setup using vLLM with prefix caching enabled. For TRIM-Thr, GPU memory was allocated as follows: the large model $M_s$ used both GPUs with tensor parallelism (0.55 memory utilization per GPU), while the small model $M_w$ and the PRM each ran on separate GPUs (0.4 utilization each). For the single large model baselines, both GPUs were fully dedicated to serving the large model. All measurements were conducted on the MATH-500 test dataset and we evaluated TRIM-Thr with two model pair configurations:
> * TRIM-Thr (1.5B + 32B): Qwen2.5-1.5B-Instruct as $M_w$ and Qwen2.5-32B-Instruct as $M_s$
> * TRIM-Thr (1.5B + 7B): Qwen2.5-1.5B-Instruct as $M_w$ and Qwen2.5-7B-Instruct as $M_s$
>
>
> | Model Configuration | Threshold (k) | Latency (sec/query) | Throughput (tok/sec) |
> |---------------|------|-------------------|-------------------|
> | Baselines | | | |
> | Qwen2.5-32B | — | 17.10 | 31.77 |
> | Qwen2.5-7B | — | 10.27 | 65.20 |
> | | | | |
> | TRIM-Thr (1.5B + 32B) | | | |
> | | 0.1 | 6.21 | 64.30 |
> | | 0.4 | 9.02 | 52.30 |
> | | 0.7 | 12.10 | 47.86 |
> | | | | |
> | TRIM-Thr (1.5B + 7B) | | | |
> | | 0.1 | 6.35 | 66.66 |
> | | 0.4 | 8.17 | 65.25 |
> | | 0.7 | 9.51 | 64.61 |
>
> These results demonstrate that TRIM-Thr (1.5B + 32B) is 1.4×–2.75× faster (17.10 / 12.10 $\approx$1.41, 17.10 / 6.21 $\approx$2.75) than running the large 32B model alone, even though TRIM involves PRM evaluation and stepwise routing. Furthermore, we observe that the latency improvements for TRIM increase as the model size grows and the threshold decreases.
>
>
> **Theoretical Justification:** The small, cheap LLM $M_w$ serves as the draft model, while the strong model maintains a shadow prefill (chunked KV-cache) of the ongoing $M_w$ generation. As a result, routing decisions do not introduce sequential stalls: when escalation occurs, the strong model can immediately continue decoding from its cached prefix.
>
> Furthermore, in **all three TRIM approaches (TRIM-Thr, TRIM-Agg, TRIM-POMDP), the routing decision computation time can be made negligible**.
> * TRIM-Agg uses a small MLP with two hidden layers (128 units each), yielding effectively zero inference overhead relative to LLM decoding time.
> * For TRIM-POMDP, we use SARSOP, an offline POMDP solver. To achieve the best performance out of the POMDP solver, we recompute the policy at every routing decision by initializing the initial state distribution with the current belief distribution (one can skip recomputation when belief changes are small without any loss in performance). However, because the true state space is small and each policy computation is fast (~5 seconds), this enables us to **precompute a lookup table mapping belief distributions to actions**, resulting in zero latency from the router during inference. Instead of this, we can also apply additional heuristics to avoid policy recomputation at every step, which work well in practice and retain optimal performance (Line 739-742).
>
> With such an implementation, because large-model decode tokens dominate wall-clock time and TRIM substantially reduces the number of these tokens, the overall inference becomes faster than running the expensive model alone.

---

> ### Author Response · Authors · 2025-11-22
> **Rebuttal by Authors - II**
>
> ## Re: Regenerate Action Definition
> > Why did you choose to return control to the weak model $M_w$ immediately after an expensive $M_s$  intervention?
>
> Our choice to return control to $M_w$ after a single $M_s$ intervention is motivated by an empirical observation (also supported by [1,2]): when an incorrect step causes the reasoning trajectory to diverge, correcting just that critical step is often sufficient to steer the solution back onto a successful path. In our experiments, we observed that when $M_w$ produces an incorrect step, the primary issue is not that all subsequent steps require $M_s$, but rather that the trajectory has diverged at a critical decision point. When that specific step is regenerated by $M_s$, the corrected token sequence frequently realigns the reasoning process, after which $M_w$ is able to continue successfully without additional intervention. This is empirically supported by our intervention analysis shown below, which shows that the vast majority of accuracy gains arise from just 1–3 targeted interventions, rather than sustained takeover by $M_s$. This behavior aligns with prior observations in multi-step reasoning [3,4], where a small number of pivotal steps disproportionately determine downstream correctness, and correcting these critical tokens can dramatically change the outcome.
>
> Below, we list the number of MATH-500 and AIME problems solved exclusively due to TRIM-Thr intervention (not solved by $M_w$), grouped by the number of intervention rounds.
>
> MATH-500 (500 questions)
> | TRIM-Thr Threshold | $M_s$ Rounds: 1 | $M_s$ Rounds: 2 | $M_s$ Rounds: 3 | $M_s$ Rounds: >3 |
> |-------------------|---------------|---------------|---------------|----------------|
> | 0.1 | 19 | 8 | 2 | 1 |
> | 0.35 | 30 | 17 | 4 | 7 |
> | 0.75 | 28 | 24 | 20 | 8 |
>
>
> AIME-Test (482 questions)
> | TRIM-Thr Threshold | $M_s$ Rounds: 1 | $M_s$ Rounds: 2 | $M_s$ Rounds: 3 | $M_s$ Rounds: >3 |
> |-------------------|---------------|---------------|---------------|----------------|
> | 0.1 | 14 | 9 | 5 | 0 |
> | 0.35 | 16 | 13 | 8 | 12 |
> | 0.75 | 20 | 15 | 18 | 13 |
>
> > Have you experimented with an alternative "regenerate" action where $M_s$ takes over for all remaining steps, and how does that compare on the performance-cost curve?
>
> As suggested, we additionally evaluated an alternative regenerate action in which $M_s$ takes over generation for all remaining steps. We compare TRIM-Thr under this alternative regenerate action (One-Step Thr) against the standard TRIM-Thr definition used in our experiments, and report the results below.
>
>
> | Dataset | Method | CPT(50%) | CPT(80%) | CPT(95%) | $Δ_{IBC}$ |
> |---------|---------|----------|----------|----------|-------|
> | MATH500 | | | | | |
> | | TRIM-Thr | 43.68 (9.45%) | 73.74 (15.95%) | 115.99 (25.08%) | 4.75 |
> | | One-Step Thr | 114.6 (24.78%) | 153.59 (33.22%) | 179.31 (38.78%) | 1.1 |
> | | | | | | |
> | AIME | | | | | |
> | | TRIM-Thr | 204.01 (23.47%) | 314.7 (36.2%) | 372.79 (42.89%) | 1.81 |
> | | One-Step Thr | 217.7 (25.04%) | 482.48 (55.5%) | 590.83 (67.97%) | 0.84 |
> | | | | | | |
> | OlympiadBench | | | | | |
> | | TRIM-Thr | 136.64 (20.45%) | 220.70 (33.03%) | 313.89 (46.97%) | 1.31 |
> | | One-Step Thr | 199.1 (29.79%) | 348.12 (52.09%) | 442.12 (66.16%) | 0.63 |
> | | | | | | |
> | MinervaMath | | | | | |
> | | TRIM-Thr | 65.15 (15.2%) | 92.78 (21.65%) | 148.55 (34.66%) | 2.23 |
> | | One-Step Thr | 144.86 (33.8%) | 189.97 (44.32%) | 259.03 (60.43%) | 0.5 |
>
>
> This comparison shows us that **allowing the large model to take over the remaining steps is substantially less efficient**, requiring far more expensive model tokens to reach the same performance targets.
>
>
>
> [1] Yu, Jiahao, et al. "Gpo: Learning from critical steps to improve llm reasoning." arXiv preprint arXiv:2509.16456 (2025).
>
> [2] Lin, Zicheng, et al. "Critical Tokens Matter: Token-Level Contrastive Estimation Enhances LLM's Reasoning Capability." arXiv preprint arXiv:2411.19943 (2024).
>
> [3] Bigelow, Eric, et al. "Forking paths in neural text generation." arXiv preprint arXiv:2412.07961 (2024).
>
> [4] Wang, Shenzhi, et al. "Beyond the 80/20 rule: High-entropy minority tokens drive effective reinforcement learning for llm reasoning." arXiv preprint arXiv:2506.01939 (2025).

---

> ### Author Response · Authors · 2025-11-22
> **Rebuttal by Authors - III**
>
> ## Re: PRM Sensitivity
>
> > How sensitive are TRIM-Thr and TRIM-Agg policies to the quality of the Process Reward Model?
>
> While TRIM-POMDP explicitly learns an observation function to map the imperfect PRM scores to the hidden states, TRIM-Agg implicitly learns to model the uncertainty in the PRM estimates during RL training (Line 321-322). Thus, while the performance of TRIM-Thr can degrade using noisier PRMs, TRIM-Agg maintains robust performance.
>
> To validate this, we trained TRIM-Agg on MATH-500 using noisy PRM scores and compared its performance against TRIM-Thr on OlympiadBench, with both methods relying on the noisy PRM evaluations. Noise was introduced by applying left-padding instead of the recommended right-padding during PRM evaluation, which introduces significant noise into step-level correctness estimates.
>
> | Method | CPT(50%) | CPT(80%) | CPT(95%) | $Δ_{IBC}$ |
> |---------|----------|----------|----------|-------|
> | TRIM-Thr | 187.37 (27.63%) | 450.77 (66.46%) | 564.92 (83.29%) | 0.38 |
> | TRIM-Agg | 156.96 (23.14%) | 262.82 (38.75%) | 305.01 (44.97%) | 1.16 |
>
> The results show that TRIM-Agg maintains strong cross-dataset generalization performance despite the noisy PRM: it achieves CPT(95%) on OlympiadBench using only 45% of expensive tokens, whereas TRIM-Thr suffers significant degradation, requiring over 80% of expensive tokens to reach CPT(95%). This empirically confirms that RL-trained routing policies can also learn to be robust to PRM noise, while simple threshold-based methods cannot.
>
> ## Re: Justifying POMDP Complexity
>
> > Given the extreme complexity of the POMDP (requiring annotated data to train the observation function, plus an online solver), what is the practical justification for using it over the much simpler RL-based TRIM-Agg policy?
>
> We agree that TRIM-POMDP is more involved than TRIM-Agg, but its practical value lies in where this complexity appears. The main challenge with TRIM-Agg is that it relies on long-horizon RL with sparse terminal rewards, making training both sample-inefficient and computationally expensive. In practice, TRIM-Agg can take up to 2 days of training (on two NVIDIA A100s) to reach near its optimal performance under high cost-performance trade-off parameters. Furthermore, a separate policy must be trained for each value of the tradeoff parameter $\lambda$. This limits its flexibility in settings where practitioners need to explore or dynamically adjust accuracy-cost trade-offs.
>
> TRIM-POMDP avoids this bottleneck. The observation function is fit only once from step-level annotations paired with PRM scores, and this mapping is reused across all values of the cost-performance parameter $\lambda$; moreover, the policy computation is significantly faster (~5 seconds for a given set of POMDP parameters). Furthermore, learning the observation function itself is lightweight, given the simple definition of observation that we utilize: our reflected KDE estimator can be trained offline using a small amount of annotated data (about 500 samples for MATH-500 and 1000 samples for AIME). Also, as noted above, we can precompute a lookup table mapping belief distributions to actions from the POMDP solver, resulting in zero latency from the router during inference using TRIM-POMDP.

---

> ### Author Response · Authors · 2025-11-28
> **Did our Rebuttal Address your Concerns?**
>
> Dear Reviewer,
>
> Given that there are only four more days left in the rebuttal period, we wanted to reach out to see if our rebuttal above addressed all your concerns or if any others still remained? We are happy to discuss further and provide more evidence in the few days that are left. If all your concerns are addressed, we would be really grateful if you could acknowledge that soon.
>
> Thanks,
>
> Authors

---

### Author Response · Authors · 2025-11-22
**Updated Baseline Evaluation**

During our initial evaluation of RouteLLM, we allowed the query-level routers to fallback to the cheap model $M_w$ when the expensive model $M_s$ produced an incorrect answer, since such fallback does not increase expensive-token usage. However, to ensure consistency with standard practice in prior query-routing work, we have removed this fallback behavior. In the revised experiments, each query-level router of RouteLLM now commits to exactly one model for the entire generation and this change lowers the performance of baselines (performance of all TRIM methods remains unchanged). We have updated the results accordingly and expanded the baseline coverage in the revised version of the paper. Importantly, this modification does not affect any of our earlier conclusions: query-level routers such as those in RouteLLM still exhibit weak cross-dataset generalization, often fitting to dataset-specific characteristics rather than capturing transferable routing behavior.

---

### Author Response · Authors · 2025-12-03
**Summary of Rebuttal Updates for Area Chair (Part 1/2)**

Dear Area Chair,

Thank you for overseeing the review process. Given the reviewer-data issue and the updated policy that decisions will now rely on pre-rebuttal scores, we wanted to provide a clear summary of the reviewers' main concerns and how we addressed them during the rebuttal period.

**Summary of reviewer follow-ups and score changes**

Of the four reviewers, three (with initial scores of 4, 6, and 6) have not had a chance to follow up yet. One reviewer (Reviewer oN83) did follow up, confirmed that our rebuttal satisfactorily addressed the concerns raised in their official review, and increased their score from 2 → 4. They raised an additional concern to which we responded, but they did not have the chance to follow up further due to the issue.


**Main concerns of the reviewers**

1. **Inference Latency and Wall-Clock Cost** (Reviewers X7GU, hcjw, oN83)

   > Stepwise routing may introduce substantial latency vs. a single run of expensive model $M_s$

   This has been the most dominant concern raised by the majority of reviewers.

   We provided direct empirical evidence demonstrating that TRIM does not introduce significant wall-clock overhead—and is faster than running the expensive model $M_s$ alone.  We utilized the same system optimization techniques that enable low-latency speculative decoding in our implementation to demonstrate this.

2. **PRM Sensitivity** (Reviewer X7GU, hcjw)

   > Sensitivity of TRIM to the quality of the PRM

   We explain that modeling the routing problem as a Partially Observable MDP precisely allows TRIM-POMDP to explicitly account for inaccuracies in PRM-based step-level correctness estimates via a learned observation function over unobserved ground-truth correctness states (as also noted by Reviewer X7GU). To further demonstrate robustness, we trained and evaluated TRIM-Agg on MATH-500 using intentionally corrupted PRM scores, which confirmed that RL-training implicitly learns to accommodate uncertainty in the PRM estimates, maintaining strong performance, whereas TRIM-Thr—which relies directly on raw PRM scores—experiences a substantial degradation in performance.

3. **Choice of $\mathrm{regenerate}$ action definition** (Reviewer X7GU)

   > Why return control to the weak model $M_w$ after an $M_s$ intervention, instead of letting $M_s$ generate the remaining steps?

   We justified this choice using conclusions from prior work and our new empirical evidence showing that the vast majority of accuracy gains stem from correcting only 1–3 critical steps, not from sustained takeover by $M_s$. We also directly compared against the suggested alternative $\mathrm{regenerate}$ action and demonstrated that our definition yields substantially better performance, validating the design choice.

4. **Prior work on Stepwise Generation** (Reviewer 5mDK)

   > What is the fundamental difference between TRIM and approaches in [1–3]?

   We clarified that TRIM differs from prior stepwise methods in both objective and framework. Unlike [1–3], which aim to reduce latency or modestly improve the strong model’s accuracy in a fixed high-budget regime, TRIM optimizes a cost-aware routing objective—maximizing accuracy under an explicit budget of expensive-model tokens. TRIM additionally supports budget-dependent decision policies (e.g., RL and POMDP) that explicitly model uncertainty in correctness scores—via a learned observation function in TRIM-POMDP and implicit uncertainty modeling through RL in TRIM-Agg—rather than relying on static acceptance thresholds that assume accurate step-level signals as in [1–3]. We have now incorporated this clarification into the revised paper.

5. **Budgeted accuracy, Effectiveness of Qwen2.5-Math-PRM-7B on AIME & Model family mismatch** (Reviewer 5mDK)

   > Could the authors also report results under budgeted accuracy for better comparability? How effective is Qwen2.5-Math-PRM-7B as a PRM for AIME problems? How do the authors address inconsistencies arising from using base models from different families (e.g., Qwen vs. Claude) that may produce outputs with different formats or distributions?

   - As requested, we have now added budgeted accuracy evaluations for all routing approaches in Appendix C.

   - We provided empirical evidence that our PRM was reasonably effective on AIME by comparing its AUC-ROC values with those on OlympiadBench and MATH-500.

   - Finally, we clarified that TRIM is model-pair agnostic, as it only requires that the chosen models support conditional continuation from each other’s partial responses. To substantiate this, we demonstrated strong TRIM-Thr performance using a completely different $(M_w, M_s)$ pair—across all math benchmarks and on GPQA-DIAMOND—showing that the framework remains robust despite model-family differences.

---

> ### Author Response · Authors · 2025-12-03
> **Summary of Rebuttal Updates for Area Chair (Part 2/2)**
>
> 6.  **Domain Generalization** (Reviewers hcjw, 5mDK; new concern by oN83)
>
>     > Methodological details appear specific to verifiable math datasets, where a reliable process reward model can be trained. Generalization of TRIM to other multi-step reasoning domains remains untested
>
>     To demonstrate broader applicability, we extend TRIM-Thr beyond mathematics and evaluate it on GPQA-DIAMOND—a challenging benchmark covering biology, physics, and chemistry—showing that TRIM continues to yield meaningful efficiency gains outside the math domain.
>
> 7. **Baseline Coverage** (Reviewer 5mDK)
>
>    > Insufficient baseline coverage
>
>    We expanded the baseline suite by adding Smoothie [5] and AutoMix [6], and clarified that the remaining reviewer suggested baselines were already present in adapted form. In particular, TRIM-Thr is the natural routing adaptation of Reward-Guided Speculative Decoding [1], and our BERT baseline is HybridLLM [4] instantiated for mathematical reasoning (replacing BART scores with groundtruth-correctness scores). Always-Small and Always-Large are already represented as anchor points in our performance–cost curves, and the $\Delta_{IBC}$ metric measures improvement over their linear interpolation. With Smoothie and AutoMix now fully benchmarked, the baseline coverage is complete.
>
> We hope this summary is helpful. Please let us know if any further clarification would assist in your decision-making. We greatly appreciate your time and consideration.
>
>
> [1] Liao, Baohao, et al. "Reward-Guided Speculative Decoding for Efficient LLM Reasoning." Forty-second International Conference on Machine Learning.
>
> [2] Pan, Rui, et al. "Specreason: Fast and accurate inference-time compute via speculative reasoning." arXiv preprint arXiv:2504.07891 (2025).
>
> [3] Liu, Yuliang, et al. "AdaptiveStep: Automatically Dividing Reasoning Step through Model Confidence." Forty-second International Conference on Machine Learning.
>
> [4] Ding, Dujian, et al. "Hybrid LLM: Cost-Efficient and Quality-Aware Query Routing." The Twelfth International Conference on Learning Representations.
>
> [5] Guha, Neel, et al. "Smoothie: Label free language model routing." Advances in Neural Information Processing Systems 37 (2024): 127645-127672.
>
> [6] Aggarwal, Pranjal, et al. "AutoMix: Automatically mixing language models." Advances in Neural Information Processing Systems 37 (2024): 131000-131034.

---

### Meta-Review · Area_Chair_nFVX · 2025-12-14

**Summary:**

The paper proposes the use of different models to generate different steps in the reasoning process, unlike current methods that assign each question entirely to a different model. The idea has sufficient novelty and was shown to be effective experimentally. The concerns of the reviewers are mostly addressed in the rebuttal.

**Reviewer Concerns:**

Reviewer X7GU questions the adequacy of the cost model and asks about latency. The rebuttal provided wall-clock time experiments. The reviewer asked about whether returning control to weak model is appropriate. The rebuttal experimentally justified the choice. The reviewer asks about sensitivity to PRM quality and the rebuttal provided experimental evidence of robustness. The reviewer asked about the complexity of using POMDP and the rebuttal pointed that precompiled policy is possible with small state space.

Reviewer hcjw asks about robustness, which the rebuttal addressed with experiments. The reviewer asks about latency, which the rebuttal addressed with experiments. The reviewer asks about other domains, and the rebuttal provided additional experiment on other datasets.

Reviewer 5mDK asks about fundamental differences with other approaches, which was clarified in rebuttal. The effectiveness of PRM score was addressed with additional experiments. The reviewer asked for evaluation at budgeted accuracy, which was provided in rebuttal. The reviewer asked for additional baseline, which the rebuttal provided. The reviewer asks about cross dataset generalization and experiments are provided in rebuttal.

Reviewer oN83. The reviewer was skeptical about performance metric, and rebuttal provided end-to-end latency experiment. The reviewer agreed to increase score to 4 in the comments and further asked about other domains. Further rebuttal provided experiments on an additional dataset.

**Reviewer Scores:**

Reviewer X7GU's concerns are mostly addressed, hence will likely keep or increase the score.
Reviewer hcjw's concerns are mostly addressed, hence will likely keep or increase the score.
Reviewer 5mDK's questions are mostly addressed, hence the reviewer may possibly increase the score from the score of 4.
Reviewer oN83 agreed to increase score to 4 in the comments.

---

### Decision · Program_Chairs · 2026-01-26

Accept (Poster)